# Chain-of-Focus Prompting: Leveraging Sequential Visual Cues to Prompt Large Autoregressive Vision Models

**Jiyang Zheng[1,2], Jialiang Shen[1], Yu Yao[1], Min Wang[3],**
**Yang Yang[3], Dadong Wang[2], Tongliang Liu[1]***
[1]Sydney AI Center, The University of Sydney
[2]CSIRO, Data61 [3]Shanghai Jiao Tong University
{jzhe5740,jshe9143}@uni.sydney.edu.au
{dadong.wang}@data61.csiro.au
{yu.yao, tongliang.liu}@sydney.edu.au

## Abstract

In-context learning (ICL) has revolutionized natural language processing by enabling models to adapt to diverse tasks with only a few illustrative examples. However, the exploration of ICL within the field of computer vision remains limited. Inspired by Chain-of-Thought (CoT) prompting in the language domain, we propose Chain-of-Focus (CoF) Prompting, which enhances vision models by enabling step-by-step visual comprehension. CoF Prompting addresses the challenges of absent logical structure in visual data by generating intermediate reasoning steps through visual saliency. Moreover, it provides a solution for creating tailored prompts from visual inputs by selecting contextually informative prompts based on query similarity and target richness. The significance of CoF prompting is demonstrated by the recent introduction of Large Autoregressive Vision Models (LAVMs), which predict downstream targets via in-context learning with pure visual input. By integrating intermediate reasoning steps into visual prompts and effectively selecting the informative ones, the LAVMs are capable of generating significantly better inferences. Extensive experiments on downstream visual understanding tasks validate the effectiveness of our proposed method for visual in-context learning.[1]

## 1 Introduction

Utilizing a pre-trained, general-purpose vision model to perform multiple downstream visual tasks with only a few illustrative examples represents a significant advancement toward artificial general intelligence. Recently, the emergence of Large Autoregressive Vision Models (LAVMs) (Bai et al., 2024; Guo et al., 2024) has presented a promising approach for achieving this unification of tasks. The principle behind this integration involves building an autoregressive model (Touvron et al., 2023a) that enables visual in-context learning (Bar et al., 2022; Zhang et al., 2023b; Wang et al., 2023a; Li et al., 2024), where given a test input and a pair of prompts containing an input image and its visualized target annotation, the vision models endeavor to recognize the visual patterns between the prompt image and its target, thereby making analogous predictions on the test image.

In the realm of large language models (LLMs), in-context learning (ICL) has been extensively studied (Dong et al., 2022). Among these approaches, Chain-of-Thought (CoT) prompting (Wei et al., 2022; Wang et al., 2022; Zhang et al., 2022b) is one of the most influential methods, significantly enhancing the predictive abilities of LLMs by introducing intermediate reasoning steps within the contextual language prompts. Given that LLMs and LAVMs share similar autoregressive architectures, we are inspired to explore whether injecting intermediate steps into visual contextual prompts can similarly unlock the capabilities of LAVMs. Building upon the principles of CoT prompting, we propose Chain-of-Focus (CoF) prompting, a novel prompting method tailored for LAVMs.

---

*Corresponding Author
[1]https://github.com/tmllab/2025_ICLR_COF

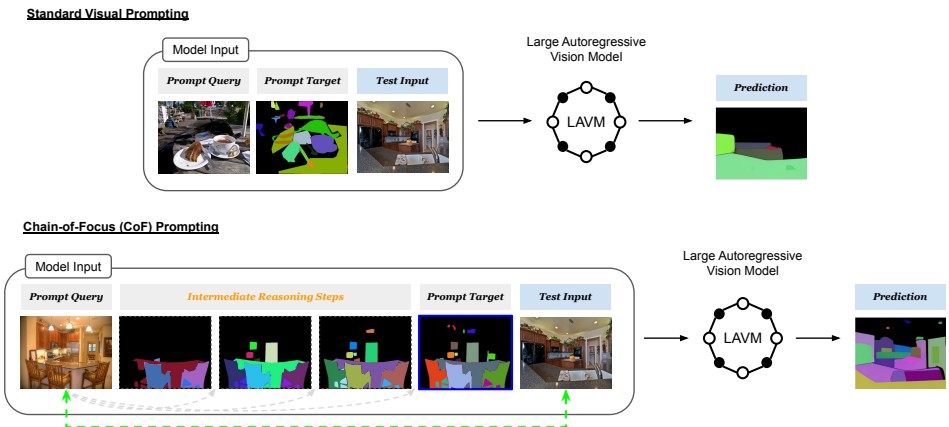

Figure 1: Illustration of Chain-of-Focus (CoF) prompting. The top section illustrates the current strategy for prompting LAVMs, where the prompt query (image) is randomly selected for the test input, and the task-specific prompt targets are visualized to form a prompt pair, enabling LAVMs to make in-context, analogy-based predictions. CoF prompting (bottom section) generates intermediate steps leading to the prompt target while selecting informative prompt pairs based on prompt query similarities to the test input and the richness of usable information contained in the prompt target.

Nevertheless, implementing contextual and sequential prompts in the vision domain presents two significant challenges. First, unlike text, which follows syntactic and semantic rules, visual data inherently lacks the clear logical structure, making it difficult to decompose and sequence for step-by-step interpretation. Second, in the language domain, hand-crafted prompts can be tailored specifically to the test input by providing analogous examples that closely relate to the problem at hand. For instance, if the test input for LLMs is a geometry problem, the language prompt can include a similar geometry problem with its solution, making the answer more informative to the model for analogy-based predictions. This level of customization is challenging in the visual domain, as images cannot be easily modified or restructured to fit new test inputs.

CoF prompting addresses the first challenge by adapting a cognitive strategy that is fundamental to human visual understanding: visual salience, which enables individuals to sequentially process visual information and draw intermediate conclusions based on the prominence of salient objects in a scene (Wertheimer & Riezler, 1944). For example, when viewing an image of a kitchen containing numerous objects, an observer's attention will initially focus on larger and closer items, such as the benchtop and chairs placed in front of it, before shifting to smaller appliances. As illustrated in Figure 1, CoF prompting replicates this cognitive process through generating intermediate reasoning steps within the prompt targets by ranking the salient regions of the prompt image in descending order. Specifically, we generate a saliency probability map using a pre-trained saliency detection model (Qin et al., 2020) to obtain the order of salient regions in the prompt image. Incrementally annotating different parts of the image based on saliency scores to create intermediate steps, allowing the models to build context progressively and enhance their predictive capabilities.

On the other hand, in the language domain, it has been shown in CoT prompting that finding informative prompt queries is crucial for enhancing LLM's predictive accuracy. Inspired by this, in CoF prompting for visual inputs, we utilize two selection criteria to search for the most informative prompts relative to the test input. First, we consider image relevance, which measures how semantically related the prompt image is to the test input image. Prior research (Zhang et al., 2023b) has demonstrated that images sharing similar semantic meanings with the test input serve as better illustrations, enabling the model to draw more accurate analogies. However, we find that for certain downstream tasks, these semantically similar images may have sparse annotations, meaning they cannot provide sufficient knowledge to the model. Therefore, we introduce the second criterion, annotation richness, to ensure that the selected prompt images contain comprehensive annotations useful for the test case. By integrating both image relevance and annotation richness, our approach addresses the challenge of creating tailored visual prompts, enhancing the model's ability to generalize from a few examples to unseen inputs.

We build our method upon the framework of Large Autoregressive Vision Models (LAVMs) (Bai et al., 2024; Hao et al., 2024), leveraging their ability to perform simultaneous predictions across multiple downstream tasks within one single pre-trained model. To quantify the similarity between the prompt image and the test image, we employ the encoder from the pre-trained LAVMs and evaluate the distance between their encoded representations. This encoder transforms raw images into discrete indices within a codebook via vector quantization(Esser et al., 2021; Van Den Oord et al., 2017). By treating these codebooks as sets and calculating the intersection over union between them, we effectively capture semantic equivalence while disregarding the specific order of indices. After identifying prompts similar to the test input, we assess the richness of prompt annotations by examining the diversity of entries in the prompt targets' codebooks. This approach ensures that the selected visual prompts are not only highly relevant but also possess rich annotations, thereby enhancing the in-context performance of the LAVMs.

To summarize our contributions, we propose a new visual prompting paradigm called Chain-of-Focus (CoF) prompting. Our approach mimics progressive thinking by incorporating intermediate steps into visual prompts and addresses the challenge of prompt customization by directly selecting the most informative prompts relative to test inputs. Our method can be seamlessly integrated with the recently proposed Large Autoregressive Vision Models (LAVMs) (Bai et al., 2024; Hao et al., 2024) through visual in-context learning, significantly improving their performance on downstream visual tasks.

## 2  RELATED WORKS

**In-Context Learning and CoT Prompting**   In-context learning (ICL) (Huang et al., 2024a; Wang et al., 2024) is a paradigm where models learn to perform tasks by conditioning on examples provided in the input context during inference. Rather than relying on traditional training processes with gradient updates, the models leverage the contextual information from query-target pairs presented at inference time to make predictions on new test inputs. In the language domain, recent advancements have highlighted the effectiveness of hierarchical reasoning techniques, known as Chain-of-Thought prompting, in enhancing the performance of large language models (LLMs) (Kojima et al., 2022; Wang et al., 2022; Wei et al., 2022; Zhang et al., 2023a; Luo et al., 2024; Zhang et al., 2024a; Lin et al., 2025; Tu et al., 2024). These methods leverage sequential reasoning steps to improve inference. Inspired by these developments, researchers have extended hierarchical reasoning frameworks to the vision-language domain (Lu et al., 2022; Zhang et al., 2023c). Among these, the most related stream of works to ours has attempted to explore the rationale within or across images and express them in textual descriptions (Ge et al., 2023; Mitra et al., 2023; Rose et al., 2023; Zheng et al., 2023). This integration of visual information into its language counterpart has yielded significant improvements for large vision-language models (LVLMs) (Liu et al., 2023; Zhang et al., 2022a; 2024b; Zhou et al., 2024), yet it also reveals the challenges of applying CoT-based methods directly to the pure vision domain (i.e., expressing reasoning without the use of language). Unlike language, images lack explicit symbolic structures, making it challenging to express reasoning steps as in LLMs or LVLMs. In purely visual contexts, Zhang et al. (Zhang et al., 2023b) develop a prompt retrieval framework for selecting in-context examples that maximize models' performance. Chain-of-Spot (Liu et al., 2024) develops a multimodal promoting method for LVLMs. It leverages language prompts to use only regions of interest (ROIs) for visual understanding. Chain-of-Sight (Huang et al., 2024b) introduces a purely visual framework that employs a sequence of visual resamplers to capture visual details at different spatial levels, generating tokens across multiple scales.

**Large Autoregressive Vision Models**   The inspiration behind autoregressive vision models stems from the advancements of large language models (LLMs) (Brown et al., 2020; Touvron et al., 2023a;b). Using contextual information, LLMs are able to capture long-range dependencies and make coherent predictions with sequential modelling techniques. Building on this concept, Bai et al. (Bai et al., 2024) propose Large Autoregressive Vision Models (LAVMs), which adapt this modelling strategy to the visual domain by constructing "visual sentences" that enable sequential prediction. This approach involves representing visual inputs as sequences of tokens, analogous to the text tokens used in LLMs. By processing visual data sequentially, the model employs self-attention mechanisms to understand dependencies and relationships within the visual context, thereby enabling effective in-context learning from purely visual inputs. By including query-target pairs from different downstream tasks in these visual sentences, the model can accomplish various visual downstream tasks within a

single framework. Hao et al. (Hao et al., 2024) extend the work and introduce a data-efficient LAVM, which is designed to operate effectively on limited datasets by making use of data augmentation and knowledge distillation. The primary purpose of LAVMs is to unify all vision tasks within a single model, making the adaptation to downstream tasks highly efficient.

# 3 METHODS

## 3.1 PRELIMINARIES

The Large Autoregressive Vision Model (LAVM) (Bai et al., 2024) is a foundational vision model that synthesizes visual predictions through sequential modeling, inspired by the successes of Large Language Models (LLMs). In LLMs, an autoregressive model predicts the next word in a sentence based on previous words. Similarly, LAVM aims to predict the next visual token in a visual sequence given the previous tokens. This is achieved using a tokenization network $E : \mathbb{R}^{h \times w \times c} \to \mathbb{R}^{n \times d}$ that transforms raw images $X = \{x_1, x_2, \ldots, x_n\} \in \mathbb{R}^{h \times w \times c}$ into visual tokens $Z = \{z_1, z_2, \ldots, z_n\} \in \mathbb{R}^{n \times d}$, followed by a sequential model $f : \mathbb{R}^p \to \mathbb{R}$ that predicts outputs in an autoregressive manner $z_t = f(z_{t-1}, z_{t-2}, \ldots, z_{t-p}) + \varepsilon_t$, where $p$ is the total number of previous time steps in the sequence, $t$ is the current step, and $\varepsilon$ is the noise. The predictions are then detokenized back to pixel space by a decoder network $D : \mathbb{R}^{n \times d} \to \mathbb{R}^{h \times w \times c}$.

In implementations of LAVMs (Bai et al., 2024; Hao et al., 2024), a pre-trained VQ-GAN (Esser et al., 2021) model is employed as the tokenizer. The VQ-GAN model encodes the image into a discrete codebook, with the indices in the codebook serving as the tokens for the autoregressive model. The pre-trained VQ-GAN decoder then decodes the codebook/tokens back into pixel space for generating images. At its core, the autoregressive model in LAVM utilizes a causal transformer (Touvron et al., 2023a) which employs causal masking to compute each token's representation based solely on itself and the preceding tokens, thereby preserving the sequence's temporal order. This allows the model to capture dependencies and patterns within the data effectively, enhancing its ability to generate coherent sequences during inference.

The visual sentences used to train LAVM are either derived from natural visual sequences, such as videos or multi-views of a 3D object (Zhan et al., 2022), or handcrafted by connecting raw images with their target annotation pairs from various visual downstream tasks. This allows the model to adapt to any downstream task given images (a.k.a. prompt queries $x_{pq}$) and annotations (a.k.a. prompt targets $x_{pt}$). At the inference stage, LAVM employs prompted inference. Given several examples of image and target annotation pairs, the tokenizer first transforms each input into tokens and constructs a visual sentence using the paired image and annotation data. The test input is appended at the end of the visual sentence as the last token. This sentence is then passed into the autoregressive network for the prediction of the next token in the sequence. The predicted tokens are subsequently constructed into a codebook and decoded into pixel space.

## 3.2 SALIENCY-BASED INTERMEDIATE REASONING STEPS

**Sequential Prompt Construction** In our approach, we construct prompts that not only present visual queries and targets but also sequentially introduce reasoning steps. Each prompt consists of a visual query $x_{pq}$, and a series of $m$ intermediate reasoning steps $\{x_{pt}^1, x_{pt}^2, \ldots, x_{pt}^m\}$ leading up to the final target $x_{pt}$. This setup mimics human reasoning processes, where intermediate conclusions are drawn before reaching a final decision. In practice, when constructing the model input with the intermediate steps, we find that the best order is denoted as: $[x_{pq}, x_{pt}^1, x_{pq}, x_{pt}^2, \ldots, x_{pq}, x_{pt}^m, x_{tq}]$, where the prompt query and intermediate targets are ordered alternately, with the test query $x_{tq}$ appended at the end of the sequence. We suggest that the optimal construction order depends on the pre-trained model itself. The pre-trained LAVMs (Hao et al., 2024; Bai et al., 2024) are primarily trained on visual sentences in the format of query and target pairs, thus its sequential prediction ability is restricted to paired representations. Similarly, finetuning the pre-trained LAVMs with natural reasoning steps allows for a different construction of prompts: $[x_{pq}, x_{pt}^1, x_{pt}^2, \ldots, x_{pt}^m, x_{tq}]$, which follows the natural sequential order. These intermediate reasoning steps decompose the complex answers into sub-pieces for understanding.

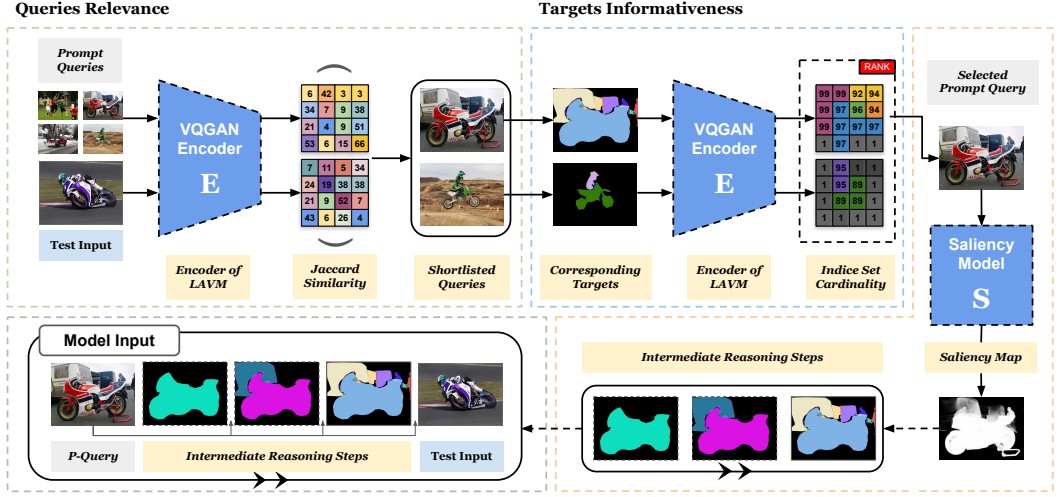

Figure 2: Illustration of Generating CoF Prompts. The framework can be viewed in two steps. First, CoF identifies a set of the most relevant queries to the test input and assesses the informativeness of their targets to filter out less informative prompt pairs. This step ensures that the prompts are highly relevant and informative to the test input. In the second step, CoF uses a saliency-based strategy to create intermediate steps for the answers to the query, which implicitly injects sequential visual cues into the prompt targets. CoF follows the general structure of Chain-of-Focus prompting, with improvements in automating the process of both prompt selection and intermediate steps generation.

**Visual Reasoning via Exploring Salient Regions**   To simulate a cognitive reasoning process, we generate a sequence of intermediate answers using visual saliency information. Given a visual query $x_{pq}$ and its corresponding answer $x_{pt}$, where both $x_{pq}$ and $x_{pt}$ are images. We utilize the salient regions within these images for constructing informative prompts. To quantitatively assess the salience of different regions within the images, we compute a saliency score $\sigma(r)$ for each region $r$, where the regions are defined by the masks on objects of interest in the image. In tasks such as image segmentation and pose estimation, the auxiliary information on masks are often provided with the ground truth as their segmentation masks and bounding boxes. We use a pre-trained saliency detection model (Qin et al., 2020) to obtain a saliency probability map for the image. For each region, we compute the saliency score as:

$$\sigma(r) = \sum_{i,j} M_r(i,j) \cdot S(x_{pq}). \tag{1}$$

$M_r(i,j)$ is the mask for the region $r$, where $M_r(i,j) = 1$ if the pixel $(i,j)$ is within the region $r$ and $M_r(i,j) = 0$ for pixels that do not belong to the region. The function $S : \mathbb{R}^{h \times w \times c} \to \mathbb{R}^{h \times w}$ extracts the pixel-wise saliency probability scores from the prompt query $x_{pq}$ and forms the probability map. The function $\sigma(r)$ computes the summed probability for the masked area as the region saliency score. We label the regions in an incremental manner using the saliency scores. For each intermediate step, the target is given by:

$$x_{pt}^{t+1} = x_{pt}^t \cup \{r \mid \sigma(r) > \tau_{t+1}\}, \tag{2}$$

where $\tau_t$ is a saliency threshold for step $t$, defining the minimum saliency required for a region to be included in the intermediate target $x_{pt}^t$. In the last step, all regions in the image will be labelled, providing a complete prompt target. This ordered introduction of information helps the LAVM to focus on relevant features at each step, allowing it to build context progressively. By leveraging saliency-based cognitive pathway, we aim to mimic the hierarchy focusing observed in human visual attention, enhancing the understanding of visual content through structured, human-like reasoning.

### 3.3   INFORMATIVE VISUAL PROMPTS

In chain-of-thought (CoT) prompting, selecting relevant queries is crucial as it directly impacts the quality of the generated responses. Traditionally, CoT involves manually choosing prompt queries

for each test input, a process that ensures alignment with desired outcomes but is labor-intensive and prone to human bias. In our method, we aim to automate this process by selecting the most relevant and informative visual query and target pairs to the test input, thereby enhancing the in-context learning performance of LAVMs. The following details our strategy for selecting visual query and target pairs to serve as the prompts for inference.

**Selection of Relevant Queries**   Given a test query $x_{tq}$ and a candidate pool of prompt pairs consisting of prompt queries $x_{pq} \in X_{pq}$ and prompt targets $x_{pt} \in X_{pt}$, our goal is to first shortlist a subset of prompts contain queries that is similar to the test query. To this end, we employ the same VQGAN encoder from the LAVM framework to serve as the feature extractor for the prompt queries and the test query. The encoder transforms the queries into discrete codebooks $\{z_{tq}, z_{pq_1}, z_{pq_2}, \ldots, z_{pq_n}\}$. Each entry in the codebook is a discrete generative factor that corresponds to the pixel space, therefore, more aligned entries in the two codebooks of queries indicate that the two queries contain similar objects or scenes in the pixel space. Through manual testing, we find that the relative position of the objects and the number of the objects in the prompt query do not affect the performance of inference as long as the two queries are semantically aligned. Hence, we convert the codebooks into sets and measure the similarity of each encoded prompt query $z_{pq}$ and $z_{tq}$ using the Jaccard similarity index, which is defined as:

$$J(z_{tq}, z_{pq}) = \frac{|z_{tq} \cap z_{pq}|}{|z_{tq} \cup z_{pq}|}. \tag{3}$$

This measure counts the number of unique indices shared between $z_{tq}$ and $z_{pq}$ without considering the position of the indices in the codebook. The set operation also helps avoid over-representation of redundant and repeating background features that are not pertinent to the task. Through this process, we shortlist a subset of $N$ prompts that have queries similar to the given test query.

**Selection of Rich Targets**   Once we have selected the $N$ most similar queries, we need to further refine our selection for target informativeness, that is to ensure the chosen answers are providing rich information for inference. As observed in tasks such as image segmentation and keypoint detection, the presence of diverse and richly annotated segmentation masks is crucial for effective in-context learning. We quantify the informativeness of a prompt target $x_{pt}$ by assessing the diversity of its encoded discrete representation $z_{pt}$. The intuition behind this involves ranking the prompt targets based on feature richness, where prompt targets with less information tend to have fewer variations in their features. For a given prompt from the shortlisted subset, we calculate the number of unique indices in its encoded targets $z_{pt}$. Formally, we maximize the function:

$$D_k(z_{pt}) = \arg \max_{z_{pt}}^{k} |z_{pt}|, \tag{4}$$

where $|z_{pt}|$ denotes the number of unique indices in $x_{pt}$'s codebook, and the $x_{pt}$ are from the shortlisted subset. We select the top $k$ prompts with the highest number of unique indices in their target codebooks, which ensures that the selected examples contain diverse annotations with varying meanings and structures. The final selection comprises the prompts with the most relevant queries and informative targets, which serve as our baseline prompts for the following visual reasoning step.

## 4   EXPERIMENTS

In this section, we conduct evaluation on CoF prompting for LAVMs. In Section 4.1, we introduce our experiment settings, including dataset, pre-trained models, metrics, and other details. In Section 4.2, we report our main results on downstream visual tasks and present extensive quantitative and qualitative analyses. In Section 4.3, we conduct ablation experiments on the three major components in the CoF framework to study the contributions of each module and provide discussions. Due to page limitations, we have included additional results and analyses in the Appendix.

### 4.1   EXPERIMENTAL SETUP

**Tasks and Dataset**   For our experiments, we select four downstream visual tasks: image segmentation (Hong et al., 2024), object detection (Zheng et al., 2022), image inpainting and pose estimation.

| Method / Model | Image Segmenation | | | | | |
| --- | --- | --- | --- | --- | --- | --- |
| | LLaMA-300M (Hao et al., 2024) | | LLaMA-1B (Hao et al., 2024) | | LLaMA-7B (Bai et al., 2024) | |
| | IoU (%↑) | P-ACC (%↑) | IoU (%↑) | P-ACC (%↑) | IoU (%↑) | P-ACC (%↑) |
| Random Selection | 26.31 ± 0.8 | 42.96 ± 1.1 | 27.21 ± 0.4 | 41.88 ± 1.0 | 45.69 ± 1.4 | 59.06 ± 2.2 |
| SegGPT (Wang et al., 2023b) | 26.52 ± 1.4 | 42.54 ± 2.7 | 26.39 ± 1.2 | 42.71 ± 1.6 | 45.38 ± 0.8 | 60.72 ± 1.9 |
| SupPR (Zhang et al., 2023b) | 27.05 ± 1.1 | 43.52 ± 1.4 | 27.94 ± 0.9 | 42.16 ± 1.2 | 49.41 ± 1.7 | 65.04 ± 1.1 |
| CoF Prompting (Ours) | **28.35**± 0.6 | **46.36** ± 0.8 | **28.79** ± 0.3 | **44.75** ± 1.0 | **52.53** ± 0.3 | **67.05** ± 0.7 |

Table 1: Segmentation results of CoF prompting on LLaMA-300M, LLaMA-1B and LLaMA-7B.

| Method / Model | Pose Estimation | | | | | |
| --- | --- | --- | --- | --- | --- | --- |
| | LLaMA-300M (Hao et al., 2024) | | LLaMA-1B (Hao et al., 2024) | | LLaMA-7B (Bai et al., 2024) | |
| | IoU (%↑) | P-ACC (%↑) | IoU (%↑) | P-ACC (%↑) | IoU (%↑) | P-ACC (%↑) |
| Random Selection | 0.60 ± 0.07 | 1.44 ± 0.09 | 1.00 ± 0.05 | 2.96 ± 0.10 | 2.40 ± 0.07 | 10.23 ± 0.16 |
| SupPR (Zhang et al., 2023b) | 0.67 ± 0.04 | 1.65 ± 0.13 | 1.04 ± 0.02 | 2.93 ± 0.18 | **2.87** ± 0.22 | 11.29 ± 0.21 |
| CoF Prompting (Ours) | **0.68** ± 0.04 | **1.75** ± 0.05 | **1.09** ± 0.02 | **3.29**± 0.07 | 2.80 ± 0.04 | **13.34** ± 0.13 |

Table 2: Pose Estimation Results of CoF Prompting on LLaMA-300M, LLaMA-1B and LLaMA-7B.

Image segmentation involves partitioning an image into multiple segments or regions. The primary objective of this task is to label each pixel in the image with a class label, identifying the object to which it belongs. Pose estimation refers to the task of determining the configuration of the body in a given image by predicting the locations of keypoints or joints. The goal here is to detect and classify the keypoints representing the positions of body parts. To facilitate these tasks, we employ the MS-COCO dataset (Lin et al., 2014), adhering to the settings outlined in Bai et al. (2024); Guo et al. (2024). Our experimental protocol involves extracting 50,000 training images and their corresponding target annotations to form the candidate prompt pool, and we rigorously test our methods on the entire validation dataset. Note that, the pre-trained LLaMA-300M and LLaMA-1B only support the image segmentation and pose estimation tasks, while LLaMA-7B supports all four downstream tasks.

**Pre-trained Models** We utilize pre-trained LAVMs from (Bai et al., 2024) and (Hao et al., 2024) for in-context learning. Specifically, we employ the VQ-GAN model as proposed by (Chang et al., 2023) to generate discrete visual representations of 2048 dimensions. For the autoregressive network, we leverage pre-trained LLaMA models (Touvron et al., 2023a;b) at different scales, including LLaMA-300M, LLaMA-1B, and LLaMA-7B for sequence modeling. Additionally, we incorporate an off-the-shelf saliency detection model from $U^2$-Net (Qin et al., 2020), which takes RGB images as input and outputs a saliency probability map of the same height and width as the input image.

**Visual ICL Baselines** We compare our method with existing visual in-context learning approaches, specifically SupPR (Zhang et al., 2023b) and SegGPT (Wang et al., 2023b). SupPR is a general prompt retrieval framework that extracts prompt pairs that contain images similar to the test input. SegGPT is a prompting method designed for segmentation tasks. We only adopt its central idea of using the same color mask for the same object class when prompting for segmentation tasks.

**Post-processing and Evaluation Metrics** Following (Guo et al., 2024; Zhang et al., 2023b), We utilize Intersection over Union (IoU) and Pixel accuracy (P-ACC) as our evaluation metrics for segmentation and pose estimation. We convert the predicted outputs into binary pixel masks and compare them against the binary ground truth masks. IoU measures the overlap between the predicted and ground truth regions by dividing the area of intersection by the area of union. The P-ACC calculates the proportion of correctly classified foreground pixels in the binary prediction mask compared to the ground truth. For detection, since the model only outputs the visualised bounding box as in image, we cannot directly obtain the coordinates for evaluation. To address this, we employ a post-process network that intakes images with visualised bounding box and outputs the box coordinates. We then calculate the IoU of the bounding boxes to the ground truth. We name the metric as Learned IoU (L-IoU). For image inpainting, we report the MSE loss and LPIPS score. We also measure the failure cases of LAVMs in making predictions, that is when the LAVMs fail to output any meaningful prediction, where the output appears in pure black. Due to page limit, we put the analysis of object detection and image inpainting in the Appendix section A.

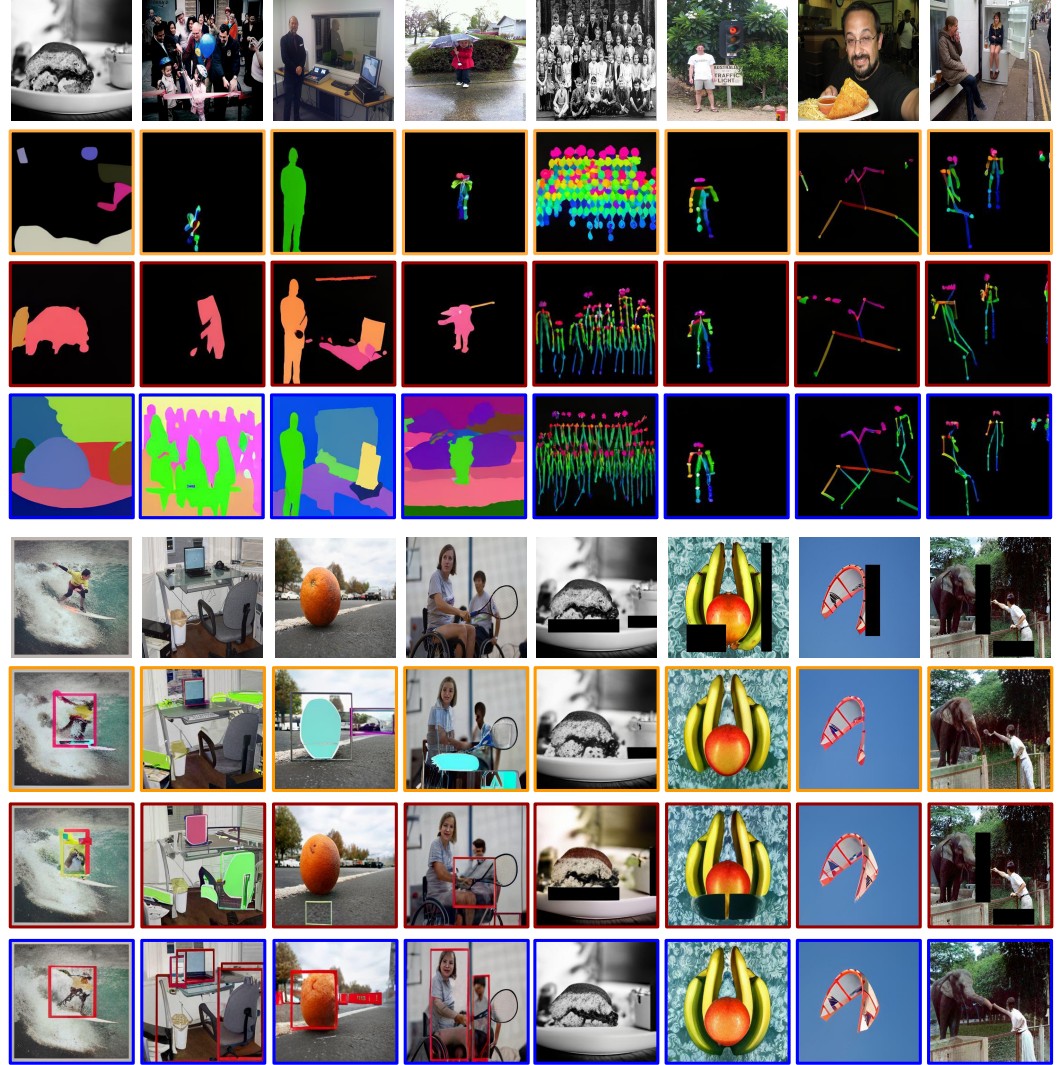

Figure 3: Results on LLaMA-7B Model. The first and fourth rows are the original test inputs for image segmentation, detection, inpainting and pose estimation, respectively. Orange boxes show the predictions given random prompts. Maroon boxes show the predictions using SupPR method (Zhang et al., 2023b). Blue boxes show the predictions using Chain-of-Focus prompting.

## 4.2 RESULTS

**Image Segmentation** Table 1 reports the quantitative performance of CoF compared to random prompting, same colour masking (Wang et al., 2023b) and SupPR (Zhang et al., 2023b). The CoF method demonstrates notable percentage increases compared to the second best performing methods across various metrics. For

| Model | Random | CoF |
|---|---|---|
| LLaMA-300M w/ VQ-GAN | 58.58 ± 1.8 | **57.62** ± 0.6 |
| LLaMA-1B w/ VQ-GAN | 50.08 ± 2.5 | **43.12** ± 1.9 |
| LLaMA-7B w/ VQ-GAN | 45.28 ± 1.1 | **42.03** ± 0.5 |

Table 3: Failure Rates (↓) - Image segmentation

image segmentation with LLaMA-300M, the increases are approximately 4.81% in IoU and 4.77% in P-ACC, while for LLaMA-1B and 7B, the increment in proportion is 3.04% and 6.31% in IoU, and 6.14% and 3.10% in P-ACC, respectively. The results are reported with predictions that have black rate > 0.2. We report the failure cases for segmentation in Table 3. Compared to the baseline, using CoF eliminates the failure cases caused by the incapability of two LAVMs by 1.64%, 13.9% and 7.7%, respectively. Figure 3 demonstrate the predictions made by Random, SupPR and CoF

| Model | CR | QR | AD | Image Segmentation | | Pose Estimation | |
|---|---|---|---|---|---|---|---|
| | | | | IoU (%↑) | P-ACC (%↑) | IoU (%↑) | P-ACC (%↑) |
| LLaMA-300M w/ VQ-GAN | ✓ | | | 27.92 | 46.17 | 0.65 | 1.63 |
| | | ✓ | | 26.32 | 42.99 | 0.59 | 1.41 |
| | | | ✓ | 26.13 | 41.92 | 0.61 | 1.44 |
| | ✓ | ✓ | | 28.13 | 45.32 | 0.68 | 1.77 |
| | | ✓ | ✓ | 26.95 | 41.72 | 0.63 | 1.55 |
| | ✓ | | ✓ | 28.21 | 46.10 | 0.65 | 1.71 |
| LLaMA-1B w/ VQ-GAN | ✓ | | | 28.63 | 45.07 | 1.04 | 2.87 |
| | | ✓ | | 26.14 | 43.05 | 0.99 | 2.73 |
| | | | ✓ | 27.39 | 43.02 | 1.01 | 2.84 |
| | ✓ | ✓ | | 27.90 | 44.16 | 1.09 | 3.30 |
| | | ✓ | ✓ | 27.33 | 42.19 | 1.07 | 3.01 |
| | ✓ | | ✓ | 28.50 | 44.35 | 1.12 | 3.22 |
| LLaMA-7B w/ VQ-GAN | ✓ | | | 51.74 | 67.01 | 2.91 | 13.46 |
| | | ✓ | | 50.98 | 65.07 | 2.66 | 11.62 |
| | | | ✓ | 47.13 | 61.26 | 2.41 | 10.26 |
| | ✓ | ✓ | | 52.01 | 65.83 | 2.88 | 12.89 |
| | | ✓ | ✓ | 50.34 | 64.00 | 2.67 | 11.62 |
| | ✓ | | ✓ | 51.97 | 66.41 | 2.64 | 12.55 |

Table 5: Ablation Study on the three major components involved in CoF pipeline. CR represents Cognitive Reasoning, which creates intermediate reasoning steps for the prompt target. QR represents Query relevance, which measures the similarity between the prompt queries and the test input. AD is Annotation Diversity, which involves accessing the diversity of indices within the targets' codebooks.

prompting with LLaMA-7B model. We observe improvement in the objects that models successfully identified and the accuracy of masking. The models using CoF prompting also demonstrate better scene understanding ability, which outputs complete masks for the same objects. These suggest that the in-context object discovery and segmentation ability of LAVMs can be enhanced by prompting them with our method.

**Pose Estimation** A similar trend is also found in the pose estimation task, where, as illustrated in Table 2, CoF prompting outperforms the other methods by a noticeable margin. For pose estimation using LLaMA-300M, compared to the second highest scores, the increases are approximately 1.49% in IoU and 6.06% in P-ACC.

| Model | Random | CoF |
|---|---|---|
| LLaMA-300M w/ VQ-GAN | 53.49 ± 3.7 | **52.38** ± 1.5 |
| LLaMA-1B w/ VQ-GAN | 43.67 ± 2.0 | **42.54** ± 0.9 |
| LLaMA-7B w/ VQ-GAN | 41.74 ± 1.5 | **35.62** ± 1.9 |

Table 4: Failure Rates (%↓) - Pose Estimation

Moreover, LLaMA-1B shows a larger improvement, with an increase of 4.81% in IoU and 11.15% in P-ACC. For LLaMA-7B, the P-ACC is increased by 18%, but the IoU is 2.5% lower then the second highest method. The failure rates are reported in Table 4. Despite pose estimation being a challenging task for LAVMs, CoF prompting reduces the failure rate on both LLaMA-300M, LLaMA-1B and LLaMA-7B by 2.08% and 2.59%, 14.7% respectively. We qualitatively compare the results of the LLaMA-7B model in the top three rows in Figure 3. CoF prompting demonstrates better performance compared to the baselines, with improvements in the completeness of the skeletons, the accuracy of pose detection, and the number of human targets that the models successfully identify in the given test input, demonstrating the effectiveness of our method. More results are provided in the appendix.

## 4.3 ABLATION STUDIES

**Intermediate Step and Prompt Selections** To understand the impact of various components in our CoF prompting method, we conduct a series of ablation studies. Specifically, the designed experiments isolate and evaluate the contribution of individual components by systematically removing or modifying specific parts of the model and observing the resulting performance changes. Through this analysis, we seek to identify the critical factors that drive the success of our method and provide insights into potential areas for further improvement. We divide our entire framework into three parts: cognitive reasoning (CR), query relevance (QR), and annotation diversity (AD). Cognitive reasoning involves generating intermediate reasoning steps using object saliency. When removing

CR, we directly prompt the LAVMs using the query and its complete target. Query relevance involves selecting prompts by measuring their relevance to the test input. When removing QR, we randomly sample the candidate set of prompts. Annotation diversity involves evaluating the prompt target. When it is removed, the CoF does not access the prompt target for prompt selection.

We present the results of our ablation experiments in Table 5. In the image segmentation task, for all models, we observe that the primary performance improvement originates from cognitive reasoning, which incorporates intermediate steps for the prompt targets. The standalone performance of the prompt retrieval component does not significantly benefit the LAVMs, as evidenced by the predictive performance, which is comparable to the random selection baselines. However, the integration of prompt selection with cognitive reasoning shows a marked improvement, with both CR + QR and CR + AD combinations achieving better results than cognitive reasoning alone. A similar trend is observed in pose estimation, where cognitive reasoning remains the most crucial component, demonstrating a significant enhancement when applied. Notably, in pose estimation, prompt selection can also achieve good performance independently, without the aid of cognitive reasoning. This provides insight into the contribution of each component within the framework, highlighting cognitive reasoning as the most critical strategy, with the two steps involved in prompt selection seamlessly enhancing the efficacy of the reasoning strategy.

**Number of Reasoning Steps** Here we exam the influence of different number of reasoning steps for CoF prompting. Our setting includes using $[0, 1, 2]$ intermediate steps in between the prompt queries and the prompt target. Due to the maximum input length to the autoregressive model employed in (Hao et al., 2024), injects two intermediate steps before the final targets is the maximum for in-context learning using the model.

We use the same prompt queries and original target for all three experiments to avoid influence from the prompt selection. Figure 4 demonstrates our results, where both models show improved performance with an increasing number of reasoning steps, indicating that more reasoning steps enhance their capabilities. However, in the case of the 300M model, the scores decrease when increasing from one intermediate step to two intermediate steps in image segmentation. Conversely, the LLaMA-1B model exhibits a more stable linear increment compared to LLaMA-300M, demonstrating that the larger model benefits more significantly from reasoning steps. These results highlight the importance of CoF prompting in achieving better performance.

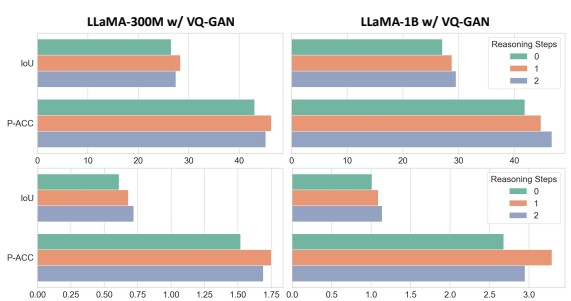

Figure 4: Comparison of using different reasoning steps. The first row of figures captures the performance measures of the image segmentation task, and the second row captures the performance measures of the pose estimation task.

## 5 CONCLUSION

The paper introduces Chain-of-Focus (CoF) prompting, a novel method designed to replicate the sequential steps of Chain-of-Thought prompting in the visual domain by bridging the gap between symbolic reasoning in language models and perceptual reasoning in vision models. CoF automates prompt design by selecting the most relevant and informative prompts from existing candidates and addresses the inherent challenge of the lack of explicit symbolic structure in images by utilizing visual saliency to create intermediate reasoning steps for prompt targets, capturing the intrinsic logic of the human perceptual system. By leveraging this hierarchical information, COF allows Large Autoregressive Vision Models (LAVMs) to process and understand visual information progressively, thus enhancing their sequential predictive performance on various downstream vision tasks. Our experiments on image segmentation and pose estimation using LLaMA-300M, 1B and 7B w/ VQ-GAN models demonstrate that embedding visual reasoning into prompts significantly improves the model's inference capabilities. CoF prompting represents a significant advancement in visual in-context learning, with potential for broader applications in machine learning and computer vision.

## 6 ACKNOWLEDGEMENT

The authors would like to thank Muyang Li for his valuable feedback throughout the project. Jiyang Zheng is supported by the CSIRO Next Generation Graduates and AI for Missions PhD program. Tongliang Liu is partially supported by the following Australian Research Council projects: FT220100318, DP220102121, LP220100527, LP220200949 , IC190100031.

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

## A ANALYSIS OF OBJECT DETECTION AND IMAGE INPAINTING

**Object Detection** Table 6 presents the quantitative performance of our CoF method compared to baselines. CoF achieve increment over the random on the L-IoU by 14.8%. While the quantitative results of SupPR and CoF are very similar in this

| Model | Random | CoF |
|---|---|---|
| LLaMA-7B w/ VQ-GAN | $57.49 \pm 3.7$ | $\mathbf{51.63} \pm 0.8$ |

Table 7: Failure Rates ($\downarrow$) - Object Detection

tasks, with CoF slightly higher in the metric. However, by observing the qualitative results in Figure 3,

| Method | Object Detection | Image Inpainting | |
|---|---|---|---|
| | L-IoU (%↑) | MSE (%↓) | LPIPS (%↓) |
| Random | 17.19 ± 0.6 | 0.91 ± 0.04 | 0.64 ± 0.01 |
| SupPR (Zhang et al., 2023b) | 19.65 ± 2.9 | 0.87 ± 0.06 | 0.55 ± 0.07 |
| CoF Prompting (Ours) | **19.74**± 0.8 | **0.61** ± 0.01 | **0.47** ± 0.02 |

Table 6: Object Detection and Image Inpainting Results of CoF Prompting on LLaMA-7B.

we can still observe the difference in between the two methods, where CoF are more accurate in locating the boxes and reconstruct the original input. We additional calcuate the failure rate, where the predicted bounding boxs are completely disjoint to the ground truth boxs. Failure cases for detection are detailed in Table 7, where CoF reduces failures by 11.9% on LLaMA-7B for object detection.

**Image Inpainting**    As shown in Figure 3, the overall qualitative performance of LAVM on inpainting task is exceptional. However, they still benefit from proper prompting. By applying CoF prompting, the generated patches are more natural and of higher quality compared to the baselines. Table 6 (Right) shows that our CoF method quantitatively outperforms the baselines, achieving a 4.3% and 1.7% improvement in MSE and a improvement in LPIPS over the second-highest prompting method.

## B    THRESHOLDING PERFORMANCE ANALYSIS

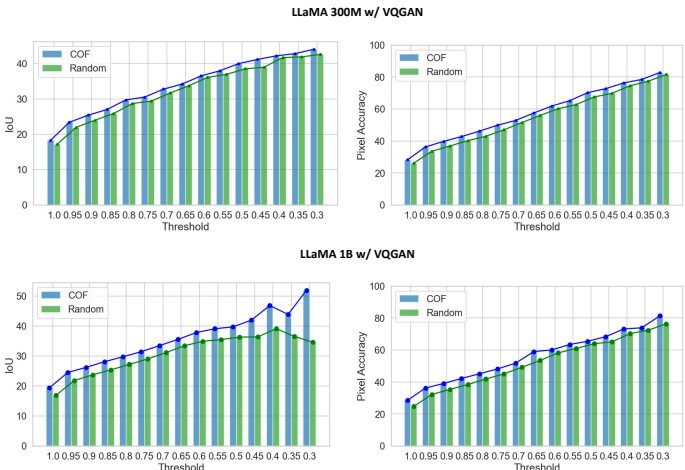

Figure 5: Image Segmentation and Pose Estimation Results for various black rate thresholding. Our method consistently outperforms the baselines on different pre-trained models across various threshold rates, demonstrating the stable performance of CoF prompting.

In this section, we analyze segmentation performance by thresholding the black rate of the prediction. The black rate represents the proportion of the black area in the predicted results. We assess the performance of COF prompting at different sizes of the predictable object to ensure its contribution is stable to the LAVMs. Figure 5 demonstrates the performance comparison between COF prompting and the Random baseline across two metrics. Across varying thresholds, COF consistently outperforms the Random baseline. This showcases that COF prompting maintains robust performance in enhancing the predictive capabilities of LAVMs regardless of the size of the predictable object.

## C    VISUALISATION OF RESULTS OF LAVM W/ LLAMA-1B

Here we present qualitative results of image segmentation and pose estimation using LAVM with LLaMA-1B. As demonstrated in Figure 6, using CoF prompting significantly improves the accuracy of object mask identification for image segmentation. Similarly, the quality of the estimated skeletons is also better when applying CoF prompting.

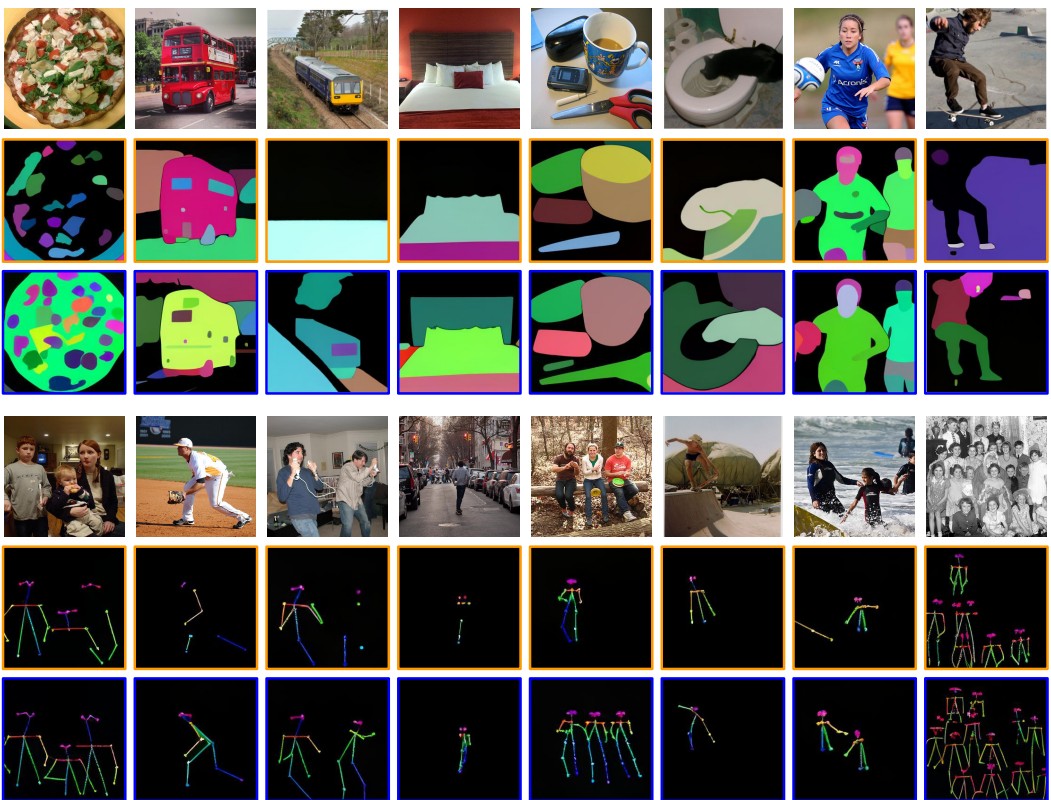

Figure 6: Results on LLaMA-1B Model. The first and fourth rows are the original test inputs for image segmentation and pose estimation, respectively. Orange boxes show the predictions given random prompts. Blue boxes show the predictions using Chain-of-Focus prompting.

## D    REVERSING ORDER OF INTERMEDIATE REASONING STEPS

| Prompting Method | LLaMA-7B | | | |
| --- | --- | --- | --- | --- |
| | Image Segmentation | | Pose Estimation | |
| | IoU (%↑) | P-ACC (%↑) | IoU (%↑) | P-ACC (%↑) |
| COF | 52.53 | 67.05 | 2.80 | 13.34 |
| COF-reversed | 49.65 | 65.37 | 2.71 | 10.52 |

Table 8: Reversed Intermediate Reasoning Steps with the LAVM w/ LLaMA-7B

The core of CoF prompting is to generate a series of intermediate reasoning steps for sequentially prompting the LAVMs. The reasoning path is created based on the saliency paths we identify within individual images. Here, we explore whether the order of reasoning steps will affect the in-context learning of LAVMs. To this end, we present a qualitative comparison in Figure 7 and a quantitative comparison in Table 8. It is observed that reversing the order of the intermediate steps can impact the in-context predictions of LAVMs; however, compared to the predictions in Figure 3, we can conclude that reverse sequential prompting is still better than directly showing the LAVMs the full target.

## E    DEPENDENCY ON SALIENCY DETECTORS

Measuring saliency (Huang et al., 2023) is an important step in our method. Here, we assess the sensitivity of our prompting method to variations in saliency detectors, we further employed a different approach: GradCAM (Selvaraju et al., 2017) to compute saliency scores. Figure 8 demonstrate the

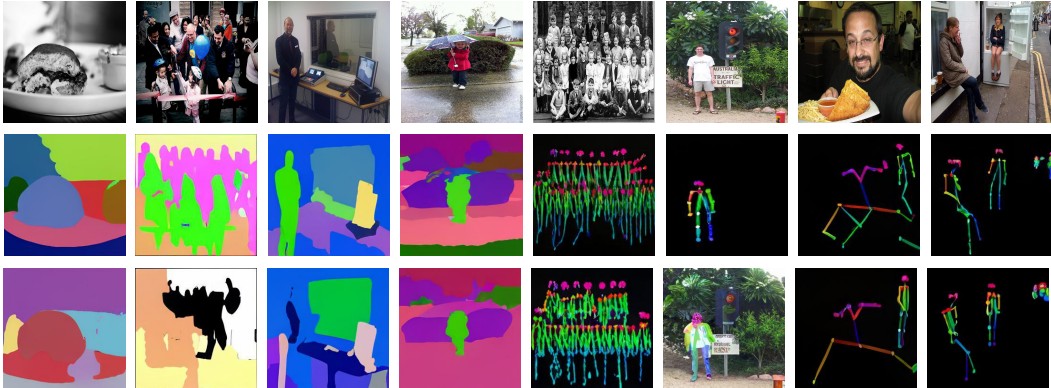

Figure 7: Qualitative Results of reversing intermediate reasoning steps with the LAVM w/ LLaMA-7B. The second row shows the CoF prompting output. The third row show the results of using the same prompt, but reversing the order in intermediate steps.

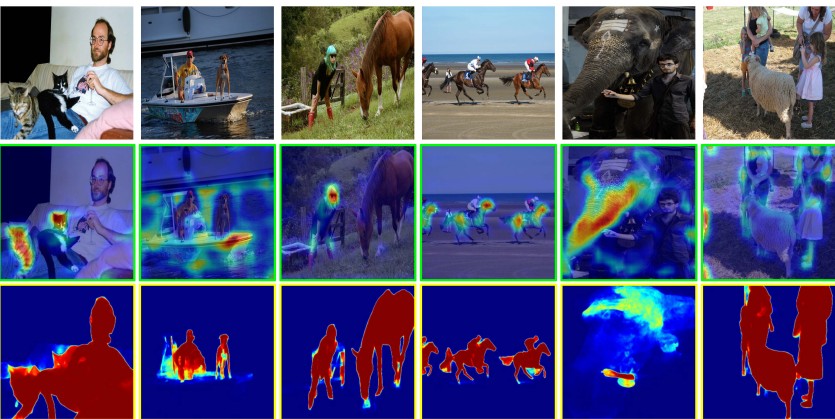

Figure 8: Visual Attention of different saliency detectors. Green boxes show the results given by GradCAM. Yellow boxes show the results given by U2-net.

different attention maps visualized from GradCAM and U2-Net, respectively. Table 9 shows the results for the LLaMA-7B LAVM on the four tasks, comparing U2-Net and GradCAM. Notably, switching the method for measuring saliency scores does not result in significant differences in performance. Based on the observation, we conclude that both approaches effectively detect salient regions, and the consistent in-context learning performance further highlights the robustness of our approach.

| Prompting Method\Task | Segmentation | Pose Estimation | Object Detection | Image Inpainting |
|---|---|---|---|---|
| | IoU (↑)/ P-ACC (↑) | IoU (↑)/ P-ACC (↑) | L-IoU (↑) | MSE (↓) / LPIPS (↓) |
| CoF Prompting w/ GradCAM | 52.68 / 67.14 | 2.77 / 13.16 | 19.77 | 0.63 / 0.51 |
| CoF Prompting w/ U2-Net | 52.53 / 67.05 | 2.80 / 13.34 | 19.74 | 0.61 / 0.47 |

Table 9: Comparison of CoF Prompting with different saliency detectors.

## F    ADDITIONAL QUALITATIVE RESULTS

In this section, we present additional visualizations of the image segmentation and pose estimation tasks to illustrate the in-context learning performance of COF prompting. For image segmentation, the results for the LLaMA-300M model are depicted in Figure 10, while Figure 11 showcases the outcomes for the LLaMA-1B model. For pose estimation, we provide visual evidence of the

improved performance facilitated by COF prompting using LLaMA-300M (Figure 12) and LLaMA-1B (Figure 13). These results demonstrate the efficacy of our method in enhancing LAVMs' predictive ability on both tasks through structured, reasoning-based prompting. Based on these additional visualizations, as well as the results shown in Figure 3, we can observe that larger models have strong predictive power on both tasks. This implies that, in order to achieve predictive capabilities similar to expert models, we will need to scale up the parameter size of the models as well as the size of the training data.

## G  LIMITATIONS AND FUTURE DIRECTIONS

While visual prompting methods can potentially enhance the predictive performance of Large Vision Models, their limitations are constrained by the capacity of these models. Practical usage of LAVMs requires stronger and more robust pretrained models, along with the advancement of in-context learning methods. Instances where current LAVMs produce pure black predictions highlight their fundamental instability, raising concerns about their trustworthiness in real-world deployment.

We observed that the failure cases are primarily associated with two factors: model scale and prompt selection. Model scale is the major factor, as LAVMs with larger parameter sizes tend to exhibit fewer failure cases. Failure cases can also arise from the choice of visual prompts, and our experiments demonstrate that the proposed prompt selection module effectively reduces the number of failures.

To further investigate, we identified prompts that previously caused failures in in-context predictions and were not encountered by the model during pre-training. We then switched the test input while using the same failure-inducing prompts, and the failure persisted across different test inputs. However, when we replaced the prompts with training samples from datasets used in the pre-training process, the success rate significantly increased. Based on these observations, we hypothesize that the root cause of failure cases is related to the model's out-of-distribution generalization ability with respect to the prompts. The model may fail to perform in-context learning if the prompts are unseen during pre-training or exhibit a domain gap.

The community as a whole desires a unified solution for all vision tasks. Therefore, the authors advocate for continued research into building robust large vision models.

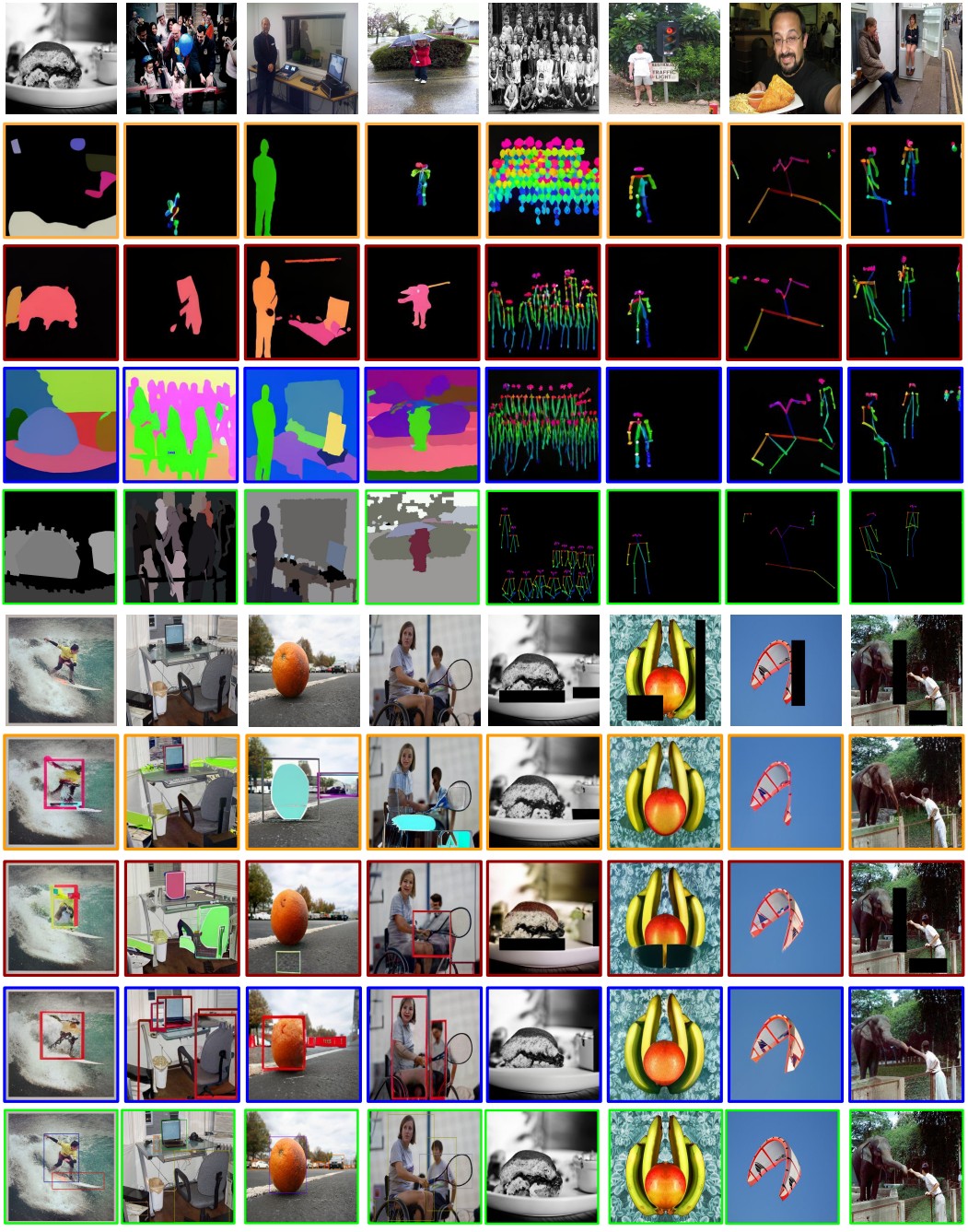

Figure 9: Ground Truth Visualization for the test input.

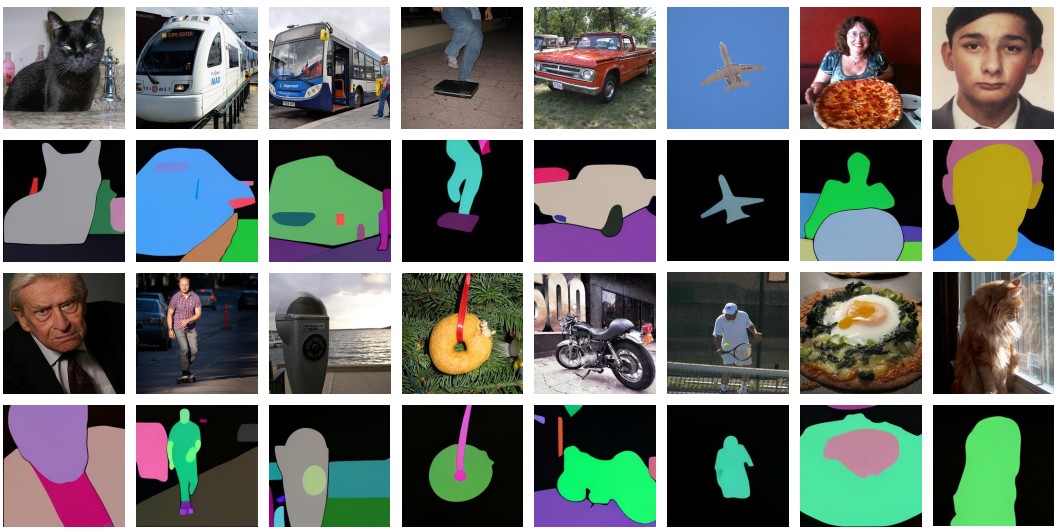

Figure 10: Image Segmentation Results from LLaMA-300M w/ VQ-GAN using COF prompting.

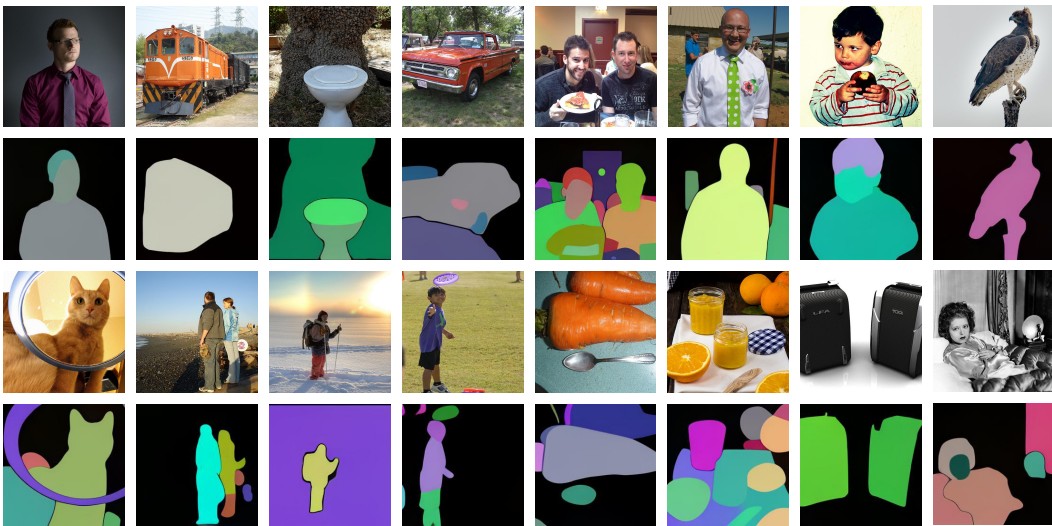

Figure 11: Image Segmentation Results from LLaMA-1B w/ VQ-GAN using CoF prompting.

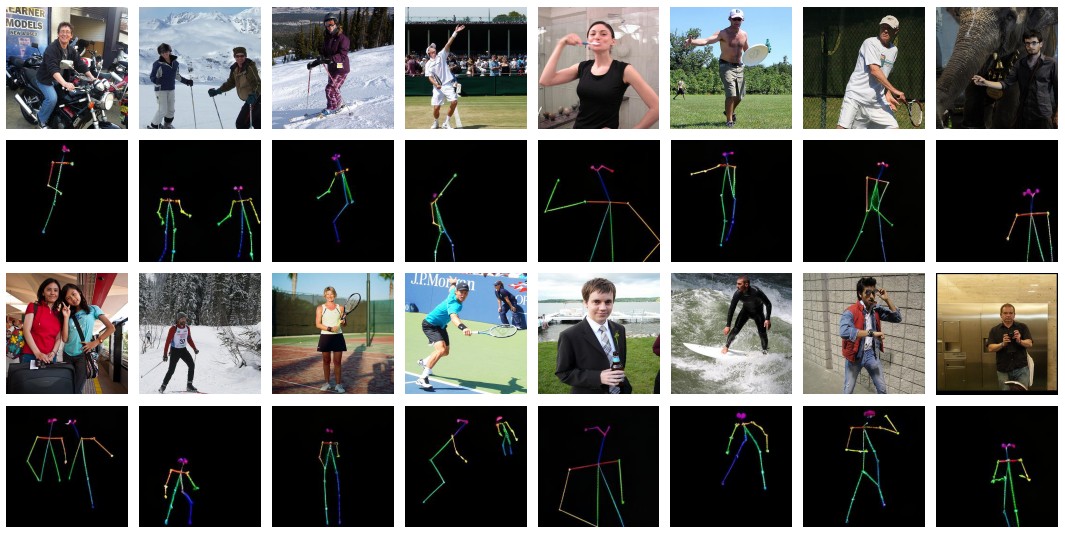

Figure 12: Pose Estimation Results from LLaMA-300M w/ VQ-GAN using COF prompting.

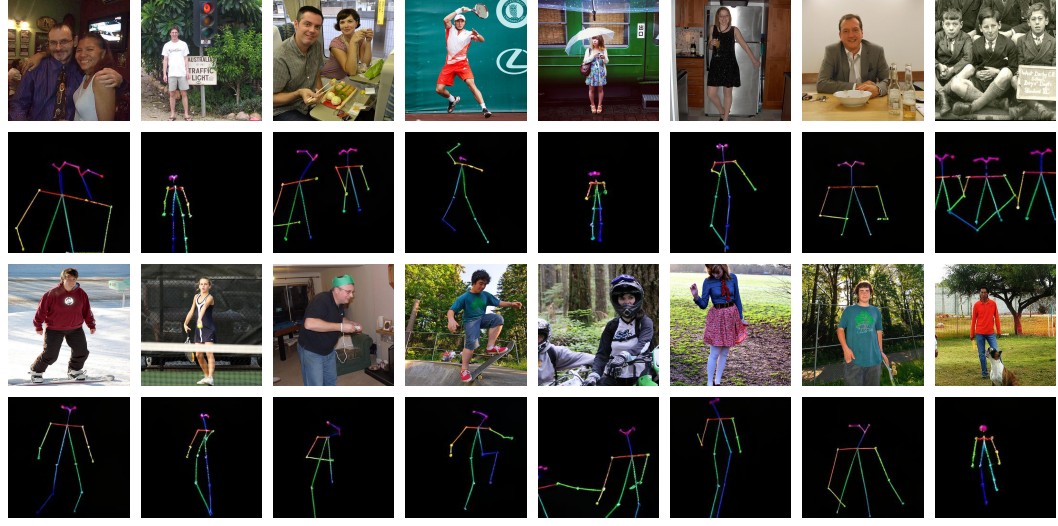

Figure 13: Pose Estimation Results from LLaMA-1B w/ VQ-GAN using CoF prompting.

