# OpenReview forum: "Chain-of-Focus Prompting: Leveraging Sequential Visual Cues to Prompt Large Autoregressive Vision Models"
_ICLR.cc/2025/Conference — ICLR 2025 Poster_

### Official Review · Reviewer_p93y · 2024-10-28

**Soundness:** 3
**Presentation:** 3
**Contribution:** 3
**Rating:** 6
**Confidence:** 4

**Summary:**

This paper aims to address an inherent challenge in the computer vision community, where, unlike language data, images lack clear logical structures. CoF prompting introduces intermediate steps in the visual learning process to enhance the ability of VLMs to understand and predict complex visual images. The authors propose using saliency maps to determine which parts of an image should be focused on sequentially, mimicking human cognitive processes. This method allows VLMs to perform better on fundamental vision tasks such as image segmentation, pose estimation, and object detection by progressively building a context around the test input. The authors comprehensively evaluated the proposed method across several models and datasets, with extensive experiments showing the efficacy of CoF prompting. It outperforms several baseline methods and demonstrates some improvements, such as Intersection over Union (IoU) and Pixel Accuracy (P-ACC). This paper also delves into ablation studies to assess the contributions of different components, providing insights into the role of cognitive reasoning, query relevance, and annotation diversity.

**Strengths:**

1. Innovative Adaptation of CoT for Vision: The introduction of Chain-of-Focus (CoF) prompting is a creative adaptation of chain-based reasoning, typically used in language models, for vision tasks. The authors contribute a novel and interesting insight into the inherent logical structures of images. Previously, we thought images contained visual redundancy, which was proposed and widely adopted in Masked Autoencoders pretraining. Such insights propose a new idea for visual modeling, which improves vision models to handle complex, multi-step tasks.

2. Comprehensive Evaluation: The authors conducted thorough experiments across various models (e.g., LLaMA-300M, LLaMA-1B, and LLaMA-7B) and tasks (e.g., segmentation, pose estimation, object detection). This robust evaluation not only demonstrates the generalizability of the approach but also its effectiveness across different settings.

3. Interesting Saliency-Based Reasoning: The use of visual saliency to create intermediate reasoning steps adds a cognitive layer to how models process visual data, mirroring human visual attention mechanisms. This results in more informed predictions, especially in tasks where spatial and object importance matters.

4. Detailed Ablation Studies: The authors provide detailed ablation studies, which highlight the relative importance of different components of the proposed method (cognitive reasoning, query relevance, and annotation diversity). This adds clarity to which parts of the model drive the most significant improvements.

**Weaknesses:**

1. Limited Exploration of Computational Complexity: While the method is novel, there is limited discussion on the computational overhead introduced by the saliency-based intermediate reasoning steps. Given that real-world applications demand efficient processing, an evaluation of the method’s scalability in terms of computational cost would have been valuable.

2. Generalization Across Different Vision Tasks: While CoF prompting is tested on several vision tasks, the experiments are focused on a few select tasks, mainly segmentation and pose estimation. Broader evaluation on a wider range of tasks, such as video understanding or 3D object recognition, would further validate the method’s versatility. Therefore, I strongly suggest the authors try some experiments on video understanding to further elaborate on the effectiveness and commonality of this method.

3. Handling Failure Cases: The authors acknowledge that VLMss can produce failures (such as pure black predictions), but the analysis of these failures remains surface-level. A deeper dive into understanding why these failures occur and how to mitigate them would strengthen the paper’s conclusions. This part could be integrated into the Conclusion and Limitation section, and it will also inspire follow-up research.

4. Dependence on Saliency Detection: The success of CoF prompting is tied closely to the quality of the saliency detection model used. The authors briefly mention the model (U2-Net), but the sensitivity of the approach to variations in saliency detection quality is not explored. If the saliency model performs poorly, it could compromise the entire prompting process.

**Questions:**

Please see the weakness.

---

> ### Author Response · Authors · 2024-11-24
> **Response to Reviewer p39y (Part 1, Q1 and Q2)**
>
> > We thank Reviewer p39y for their thoughtful comments and valuable suggestions, which have greatly contributed to enhancing the clarity and depth of our work. Below, we address the questions and comments in detail.
>
> ---
>
> > **Q1: Computational complexity**
>
> > A1: We agree that computational complexity is an important aspect of deploying methods in real-world applications. Hence, we have conducted studies on the time complexity and resource overhead for generating CoF prompts and using CoF prompting with LAVMs, the batch size is set to one. Notably, CoF prompting does not significantly increase inference time compared to the vanilla prompting method (+0.25s/per sample for the 300M model, +0.17s/per sample for the 1B model and +0.13s/per sample for the 7B model). The primary time overhead arises from traversing the candidate prompt pool to extract embeddings. To mitigate this overhead, we implemented a memory bank to store the embeddings of all candidate prompts. This optimization allows us to perform a simple linear lookup on the stored embeddings when identifying the best prompt. Since the encoded prompts are significantly smaller than the original images, the total size for a candidate pool of 50,000, stored in float32 format, is approximately 390 MB.
>
> > | LAVM Models       | Total Models Size (MB) | Peak Memory Usage (GB) | Avg Inference Time (s) |
> |-------------------|------------------------|------------------------|------------------------|
> | LLaMA-300M        | 527                    | 7                     | $\approx$ 11.55        |
> | LLaMA-300M w/ CoF | 705                    |11                     | $\approx$ 11.80        |
> | LLaMA-1B          | 2,180                  | 9.5                     | $\approx$ 17.11        |
> | LLaMA-1B w/ CoF   | 2,358                  | 11                     | $\approx$ 17.28        |
> | LLaMA-7B          | 13,090                 | 27                     | $\approx$ 31.37        |
> | LLaMA-7B w/ CoF   | 13,268                 | 27                     | $\approx$ 31.50        |
>
> ---
>
> >**Q2: Generalization across tasks**
>
> > A2: We appreciate the suggestion to include experiments on video understanding or 3D object recognition. While this suggestion is highly appealing, the current pre-trained LAVMs have limitations in their capabilities. At present, these models do not support tasks such as multi-object video recognition or 3D object recognition. Among the tasks that can benefit from chain reasoning, we have validated our method on four distinct tasks: segmentation, pose estimation, object detection, and image inpainting. The results have demonstrated the effectiveness of our proposed method. While CoF Prompting effectively enhances the in-context predictive ability of LAVMs, it does not enable out-of-task generalizability (i.e., performing tasks that the models were not trained for). Such generalizability remains beyond the current capabilities of LAVMs. To extend CoF Prompting to tasks like multi-object video recognition or 3D object recognition, a stronger pre-trained LAVM, with these tasks explicitly included in its training process, would be required.

---

> ### Author Response · Authors · 2024-11-24
> **Response to Reviewer p39y (Part 2, Q3 and Q4)**
>
> >**Q3: Discussion about failure cases**
>
> > A3: Thank you for the meaningful suggestion. We have included a limitations section in the appendix to address the handling of failure cases. In summary, we observed that the failure cases are primarily associated with two factors: model scale and prompt selection. Model scale is the major factor, as LAVMs with larger parameter sizes tend to exhibit fewer failure cases. Failure cases can also arise from the choice of visual prompts, and our experiments demonstrate that the proposed prompt selection module effectively reduces the number of failures.
>
> > To further investigate, we identified prompts that previously caused failures in in-context predictions and were not encountered by the model during pre-training. We then switched the test input while using the same failure-inducing prompts, and the failure persisted across different test inputs. However, when we replaced the prompts with training samples from datasets used in the pre-training process, the success rate significantly increased. Based on these observations, we hypothesize that the root cause of failure cases is related to the model's out-of-distribution generalization ability with respect to the prompts. The model may fail to perform in-context learning if the prompts are unseen during pre-training or exhibit a domain gap.
>
> > We believe these findings are intriguing and could inspire further investigations into the robustness and generalization capabilities of LAVMs.
>
> ---
>
> >**Q4: Dependence on saliency detector**
>
> > A4: Thank you for raising this insightful question. U2-Net is used in our method as an indicator of the saliency of objects in the image. To assess the sensitivity of our method to variations in saliency detectors, we further employed a different approach: GradCAM [1] to compute saliency scores. Below are the results for the LLaMA-7B LAVM on the four tasks, comparing U2-Net [2] and GradCAM.
>
> >| Prompting Method\Task    | Segmentation                         | Pose Estimation                      | Object Detection   | Image Inpainting                          |
> |--------------------------|--------------------------------------|--------------------------------------|--------------------|-------------------------------------------|
> |                          | IoU ($\uparrow$)/ P-ACC ($\uparrow$) | IoU ($\uparrow$)/ P-ACC ($\uparrow$) | L-IoU ($\uparrow$) | MSE ($\downarrow$) / LPIPS ($\downarrow$) |
> | CoF Prompting w/ GradCAM | 52.68 / 67.14                        | 2.77 / 13.16                         | 19.77              | 0.63 / 0.51                               |
> | CoF Prompting w/ U2-Net  | 52.53 / 67.05                        | 2.80 / 13.34                         | 19.74              | 0.61 / 0.47                               |
>
> >Notably, switching the method for measuring saliency scores does not result in significant differences in performance. Based on the observation, we conclude that both approaches effectively detect salient regions, and the consistent in-context learning performance further highlights the robustness of our approach. For more details, please check the Section E in the appendix.
>
> ---
>
> > Reference: [1] Selvaraju, R.R., Cogswell, M., Das, A., Vedantam, R., Parikh, D. and Batra, D., 2017. Grad-cam: Visual explanations from deep networks via gradient-based localization. In Proceedings of the IEEE international conference on computer vision (pp. 618-626). [2] Qin, X., Zhang, Z., Huang, C., Dehghan, M., Zaiane, O.R. and Jagersand, M., 2020. U2-Net: Going deeper with nested U-structure for salient object detection. Pattern recognition, 106, p.107404.

---

> > ### Comment · Reviewer_p93y · 2024-11-25
> >
> > Thanks for the response, I will keep my positive rating.

---

### Official Review · Reviewer_GMfc · 2024-10-29

**Soundness:** 3
**Presentation:** 3
**Contribution:** 3
**Rating:** 8
**Confidence:** 4

**Summary:**

This paper extends Large Autoregressive Vision Models (LAVMs) with the capability of in-context learning for visual inputs by proposing a chain-of-focus (CoF) prompting approach. The CoF decomposes a visual prompt to intermediate reasoning steps by ranking salient regions of the prompt image, as well as taking into account the visual similarity and annotation richness to the test image. The saliency scores are obtained by an off-the-shelf saliency detector. The relevance of queries is modeled by the Jaccard similarity index of the two sets of visual codebooks in VQ-GAN. The annotation richness is modeled by the number of unique indices in their codebooks. The proposed method has been evaluated on 4 vision tasks, e.g., image segmentation, object detection, image inpainting, and pose estimation, which show the effectiveness of the CoF prompting, upon two backbone LAVMs.

**Strengths:**

1) The proposed CoF prompting method is well motivated, which shall interest many ICLR attendants.
2) The proposed approaches, using the saliency score rank, the similarity of two sets of codebooks in VQ-GAN, are reasonable and easy to re-produce.
3) The evaluation is fairly convincing to show the effectiveness of the proposed method, though not as significant as I expected.

Overall, this is a descent work which may deserve to share with the community in time.

**Weaknesses:**

1) The reasoning steps of 0, 1, 2 are evaluated in Fig.4. More steps (just 2) do not show clear advantage, which is a bit disappointing to validate the proposed CoF prompting. I wonder if just using more diverse visual prompts that are also relevant with enough annotations, the performance may match or even out-perform using intermediate reasoning steps of the visual prompts?

2) The comparison baseline is a random selection scheme, which may be too simple. The performance gain over this simple baseline is not that significant as I expected.

3) The specific approaches, like the off-the-shelf saliency score and the number of unique indices in their codebooks, are somewhat simple and intuitive.

**Questions:**

Please discuss the pros and cons of using more diverse visual prompt images or more intermediate reasoning steps of one prompt image.

Any alternatives to the saliency scores to measure the reasoning steps? Some objects like faces, animals or intensive actions tend to capture visual attentions easily.

Some relevant missing references, please discuss the differences:

Chain-of-Spot: Interactive Reasoning Improves Large Vision-language Models, 2024.
Accelerating Pre-training of Multimodal LLMs via Chain-of-Sight, 2024.

What is the subtle difference between image relevancy and relevance? ll.140 => an image retrieval framework; ll.216, in Fig.2, => query relevance or relevance of the queries; => target informativeness or informativeness of task objectives? Btw: informative and informativeness are very vague terms, please define them first.

---

> ### Author Response · Authors · 2024-11-24
> **Response to Reviewer GMfc (Part 1, Q1)**
>
> > We express our gratitude to Reviewer GMfc for the insightful feedback and constructive suggestions, which have been instrumental in improving the clarity and depth of our work. The following are our responses to the questions and comments.
>
> ---
>
> > **Q1: More diverse visual prompts**
>
> > A1: This is a very insightful and important question. We hypothesize that adding more diverse and relevant visual examples with sufficient annotations can enhance a LAVM’s in-context learning performance. This is analogous to how providing extra details in a language question for Large Language Models (LLMs) can encourage them to generate more accurate answers.
>
> >The primary focus of our paper, however, lies in exploring whether adding intermediate reasoning steps to visual targets can improve the performance of LAVMs.
>
> > In response to the comments, we first conducted an experiment to validate whether using more diverse and relevant visual prompt images leads to improved performance. Subsequently, we investigated whether incorporating reasoning steps into these cases offer additional benefits.
>
> > As shown in the table below, the results confirm the hypothesis: across all four tasks, we observe a linear positive correlation between the number of selected prompts and the corresponding metric performance. Thus, incorporating more diverse and relevant visual prompts enhances the in-context learning performance of 7B LAVMs.
>
> > We further compared the in-context learning performance using the same prompt images with only one reasoning step. Our results indicate that involving intermediate reasoning steps can further boost performance. Interestingly, in some cases, the performance improvement from adding a reasoning step surpasses that of including an additional prompt image.
>
> > These two approaches: using more diverse visual prompts and leveraging intermediate reasoning steps, represent complementary research directions. The former focuses on expanding and refining the prompt query space to enhance in-context learning, while the latter emphasizes extracting information through structured reasoning steps using the prompt target.
>
> > In our work, we explored the latter approach and demonstrated that incorporating reasoning steps significantly improves the in-context predictive ability of LAVMs. Importantly, these two strategies are not mutually exclusive; rather, they complement each other and can jointly enhance the in-context learning performance of LAVMs.
>
> > | Prompt Images | Reasoning Steps | Segmentation (IoU / P-ACC) | Pose Estimation (IoU / P-ACC) | Object Detection (L-IoU) | Inpainting (MSE $\downarrow$ / LPIPS $\downarrow$) |
> |---------------|-----------------|----------------------------|-------------------------------|--------------------------|----------------------------------------------------|
> | 1             | N               | 50.29 / 64.60              | 2.66 / 12.87                  | 18.65                    | 0.79 / 0.52                                        |
> | 1             | Y               | 51.08 / 66.17              | 2.76 / 12.29                  | 19.16                    | 0.74 / 0.50                                        |
> | 2             | N               | 50.27 / 64.54              | 2.62 / 11.04                  | 18.92                    | 0.69 / 0.50                                        |
> | 2             | Y               | 51.79 / 66.82              | 2.78 / 13.27                  | 19.71                    | 0.60 / 0.45                                        |
> | 3             | N               | 52.57 / 68.16              | 2.74 / 12.14                  | 19.25                    | 0.59 / 0.45                                        |
> | 3             | Y               | 54.19 /68.73               | 2.81 / 12.87                  | 19.77                    | 0.59 / 0.42                                        |
> | 4             | N               | 54.28 / 68.59              | 2.92 / 13.10                  | 19.73                    | 0.56 / 0.44                                        |
> | 4             | Y               | 54.70 / 69.04              | 2.96 / 13.77                  | 19.86                    | 0.56 / 0.40                                        |

---

> ### Author Response · Authors · 2024-11-24
> **Response to Reviewer GMfc (Part 2, Q2 and Q3)**
>
> > **Q2: Baselines and Improvement**
>
> > A2: We thank the reviewer for raising this question. Given the early stage of development of LAVMs, there are limited applicable in-context learning methods available for pure visual models. In addition to the random baseline, we included SupPR [1] as a baseline for all four tasks and adopted strategies inspired by SegGPT [2] to construct a baseline for segmentation. The results demonstrate that our prompting method achieves notable improvements compared to the baselines, both quantitatively and qualitatively. We acknowledge that the performance gain is less pronounced in quantitative measurements compared to qualitative observations. This discrepancy arises from the evaluation metrics used in our study. Specifically, we adopted evaluation techniques from prior work on LVMs [3,4], which primarily focus on pixel accuracy for reconstructed images. While effective for low-level accuracy assessments, these metrics are less sensitive to the scene-level improvements facilitated by CoF prompting. These improvements, however, are evident in the qualitative results.
>
> ---
>
> > **Q3: Alternative saliency measurement**
>
> > A3: Following your suggestion, we explored an alternative approach to measuring visual saliency scores by employing GradCAM [5]. We conducted a quantitative comparison between GradCAM and U2-Net [6] and observed that GradCAM captures visual attention more specifically, such as focusing on faces and certain body parts of humans (Figure 8, Appendix Section E). Subsequently, we evaluated the in-context learning performance of LAVMs when substituting U2-Net with GradCAM for saliency score computation. Notably, switching between these methods does not lead to significant differences in performance. Based on our findings, we conclude that both approaches effectively identify saliency regions for constructing reasoning steps, and the consistent in-context learning performance further highlights the robustness of our approach.
>
> >| Prompting Method\Task    | Segmentation                         | Pose Estimation                      | Object Detection   | Image Inpainting                          |
> |--------------------------|--------------------------------------|--------------------------------------|--------------------|-------------------------------------------|
> |                          | IoU ($\uparrow$)/ P-ACC ($\uparrow$) | IoU ($\uparrow$)/ P-ACC ($\uparrow$) | L-IoU ($\uparrow$) | MSE ($\downarrow$) / LPIPS ($\downarrow$) |
> | CoF Prompting w/ GradCAM | 52.68 / 67.14                        | 2.77 / 13.16                         | 19.77              | 0.63 / 0.51                               |
> | CoF Prompting w/ U2-Net  | 52.53 / 67.05                        | 2.80 / 13.34                         | 19.74              | 0.61 / 0.47                               |

---

> ### Author Response · Authors · 2024-11-24
> **Response to Reviewer GMfc (Part 3, Q4 and Q5)**
>
> ---
>
> > **Q4: Related works**
>
> > Thank you for your insightful observation. We have included discussions about the aforementioned manuscripts in the updated related work section. To summarize, Chain-of-Spot [7] is designed for Large Vision-Language Models (LVLMs). It leverages language prompts to use only regions of interest (ROIs) for visual understanding. In contrast, our method, CoF, is designed for LALMs and relies solely on visual inputs. Instead of focusing on ROIs, we emphasize the importance of providing intermediate reasoning steps in visual prompts. Chain-of-Sight [8] introduces a purely visual framework that employs a sequence of visual resamplers to capture visual details at different spatial levels, generating tokens across multiple scales. While Chain-of-Sight focuses on accelerating the pretraining of large multimodal models, CoF is specifically tailored for enhancing the in-context learning capabilities of LAVMs.
>
> ---
>
> >**Q5: Clarification of Writing**
>
> > A5: We greatly appreciate your thorough and meticulous review of our manuscript. In our context, "image relevancy" and "image relevance" have the same meaning. Target informativeness is defined as the amount of information provided by the target. If the target is sparse, then the visualized target image will contain mostly a black background (i.e., background with a few small segmentation masks), and it tends to contain less information for in-context learning. We have fixed the inconsistencies in wording and elaborated on the confusing terms in our paper. Specifically: Changed "image relevancy/relevance" to "image relevance"; Changed "An image retrieval framework" to "A prompt retrieval framework for selecting in-context examples"; Changed "Query relevance" or "relevance of the queries" to "Query Relevance"; Explained and defined "informativeness" in the method section.
>
>
> ---
>
> > Reference: [1] Zhang, Y., Zhou, K. and Liu, Z., 2023. What makes good examples for visual in-context learning?. Advances in Neural Information Processing Systems, 36, pp.17773-17794. [2] Wang, X., Zhang, X., Cao, Y., Wang, W., Shen, C. and Huang, T., 2023. Seggpt: Segmenting everything in context. arXiv preprint arXiv:2304.03284. [3] Bai, Y., Geng, X., Mangalam, K., Bar, A., Yuille, A.L., Darrell, T., Malik, J. and Efros, A.A., 2024. Sequential modeling enables scalable learning for large vision models. In Proceedings of the IEEE/CVF Conference on Computer Vision and Pattern Recognition (pp. 22861-22872). [4] Hao, Z., Guo, J., Wang, C., Tang, Y., Wu, H., Hu, H., Han, K. and Xu, C., Data-efficient Large Vision Models through Sequential Autoregression. In Forty-first International Conference on Machine Learning. [5] Selvaraju, R.R., Cogswell, M., Das, A., Vedantam, R., Parikh, D. and Batra, D., 2017. Grad-cam: Visual explanations from deep networks via gradient-based localization. In Proceedings of the IEEE international conference on computer vision (pp. 618-626). [6] Qin, X., Zhang, Z., Huang, C., Dehghan, M., Zaiane, O.R. and Jagersand, M., 2020. U2-Net: Going deeper with nested U-structure for salient object detection. Pattern recognition, 106, p.107404. [7] Liu, Z., Dong, Y., Rao, Y., Zhou, J. and Lu, J., 2024. Chain-of-Spot: Interactive Reasoning Improves Large Vision-Language Models. arXiv preprint arXiv:2403.12966. [8] Huang, Z., Ji, K., Gong, B., Qing, Z., Zhang, Q.L., Zheng, K., Wang, J., Chen, J. and Yang, M., 2024. Accelerating Pre-training of Multimodal LLMs via Chain-of-Sight. In The Thirty-eighth Annual Conference on Neural Information Processing Systems.

---

> > ### Comment · Reviewer_GMfc · 2024-11-25
> >
> > Thank you for the detailed response, which has addressed most of my concerns.
> >
> > How do you measure "the amount of information provided by the target" in your case?

---

> > > ### Author Response · Authors · 2024-11-26
> > >
> > > >Dear Reviewer GMfc,
> > >
> > > >Thank you for your feedback. We are delighted to see that our response has solved most of your concerns.
> > >
> > > >We measure the amount of information provided by the target by evaluating the diversity of its encoded discrete representation by number of unique indices in its codebook. The underlying intuition is that prompt targets with less information tend to exhibit fewer variations in their features.
> > >
> > > >Specifically, for a given prompt target $x_{pt}$, we calculate the number of unique indices in its encoded representation $z_{pt}$. For the set of candidate prompt targets, we select the most informative prompts by maximizing the following function:
> > >
> > > >$$D(z_{pt}) = \arg\max_{z_{pt} {}} |z_{pt}|$$
> > >
> > > >where $|z_{pt}|$ denotes the number of unique indices in $x_{pt}$'s codebook.

---

> > > > ### Comment · Reviewer_GMfc · 2024-12-02
> > > >
> > > > While, I would say this is a reasonable indication of the amount of information. Since most of my questions have been well addressed, and the proposed chain-of-focus idea is quite interesting to share with the community, I'd like to raise my rating. Thanks!

---

### Official Review · Reviewer_eyXR · 2024-11-02

**Soundness:** 3
**Presentation:** 3
**Contribution:** 3
**Rating:** 8
**Confidence:** 4

**Summary:**

This paper propose Chain-of-Focus(COF) Prompting, which enhances vision models by enabling step-by-step visual comprehension. CoF automates prompt design by selecting the most relevant and informative prompts from existing candidates and creates intermediate reasoning steps for prompt targets. Experiments on various downstream vision tasks demonstrate the effectiveness of the proposed method.

**Strengths:**

This paper is well written. The methods are innovative, the experimental results are significant, and the visualizations are intuitive.

**Weaknesses:**

1. The candidate prompt pool comprises 50,000 training images and annotations, which raises concerns in specific scenarios due to its inability to ensure data privacy. Additionally, it necessitates significant resources for the storage and retrieval of prompts.
2. Figure 3 does not include visualized results of the ground truth.
3. There is no comparison of the computational complexity and resource overhead of the different methods.

**Questions:**

Can CoF address the issue of distribution shift, that is, when there is no data related to the test data in the prompt pool?

---

> ### Author Response · Authors · 2024-11-24
> **Response to Reviewer eyXR**
>
> > We sincerely appreciate Reviewer eyXR for their insightful feedback and constructive suggestions, which have significantly contributed to enhancing the clarity and depth of our work. Below, we provide detailed responses to the questions and comments.
>
> ---
>
> > **Q1: Resources cost and data privacy**
>
> > A1: We conducted a comparison of the computational complexity and resource overhead for generating CoF prompts and using CoF prompting with LAVMs, the batch size is set to one. Notably, CoF prompting does not significantly increase inference time compared to the vanilla prompting method (+0.25s/per sample for the 300M model, +0.17s/per sample for the 1B model and +0.13s/per sample for the 7B model). The primary time overhead arises from traversing the candidate prompt pool to extract embeddings. To mitigate this overhead, we implemented a memory bank to store the embeddings of all candidate prompts. This optimization allows us to perform a simple linear lookup on the stored embeddings when identifying the best prompt. Since the encoded prompts are significantly smaller than the original images, the total size for a candidate pool of 50,000, stored in float32 format, is approximately 390 MB.
>
> >| LAVM Models       | Total Models Size (MB) | Peak Memory Usage (GB) | Avg Inference Time (s) |
> |-------------------|------------------------|------------------------|------------------------|
> | LLaMA-300M        | 527                    | 7                     | $\approx$ 11.55        |
> | LLaMA-300M w/ CoF | 705                    | 11                     | $\approx$ 11.80        |
> | LLaMA-1B          | 2,180                  | 9.5                     | $\approx$ 17.11        |
> | LLaMA-1B w/ CoF   | 2,358                  | 11                     | $\approx$ 17.28        |
> | LLaMA-7B          | 13,090                 | 27                     | $\approx$ 31.37        |
> | LLaMA-7B w/ CoF   | 13,268                 | 27                     | $\approx$ 31.50        |
>
> > On the other hand, creating a memory bank for the candidate pool also addresses data privacy concerns. The original data in the candidate pool can be removed after extracting the embeddings, as these embeddings can be directly utilized by the autoregressive network in LAVMs to make predictions.
>
> ---
>
> > **Q2: Distribution shift**
>
> > A2: To investigate whether CoF prompting can address the distribution shift problem, we conducted experiments using out-of-distribution (OOD) data as the test input. Specifically, we selected prompts from the COCO dataset [1] while performing in-context predictions on the Pascal VOC dataset [2] for the segmentation task. It is observed that CoF prompting achieves notably higher performance compared to other prompting methods in this OOD setting, as shown in the table below. From previous studies, it is known that LAVMs possess a degree of generalization ability towards unseen test data [3]. Based on our observations, we suggest that CoF prompting not only maintains this ability but also further enhances predictive performance on unseen data by structuring the reasoning steps effectively.
>
> >| Segmentation w/ OOD Test Data |                                      |
> |-------------------------------|--------------------------------------|
> | Prompting Method              | IoU ($\uparrow$)/ P-ACC ($\uparrow$) |
> | Random                        | 50.73 / 67.04                        |
> | SegGPT                        | 51.25 / 67.08                        |
> | SupPR                         | 53.04 / 71.57                        |
> | CoF                           | 59.10 / 74.13                        |
>
> ---
>
> > **Q3: Visualization of ground truth**
>
> > A3: Thank you for bringing this up. We have now included the ground truth and our predictive results in Figure 9 in the appendix section F.
>
> ---
>
> >Referecne: [1] Lin, T.Y., Maire, M., Belongie, S., Hays, J., Perona, P., Ramanan, D., Dollár, P. and Zitnick, C.L., 2014. Microsoft coco: Common objects in context. In Computer Vision–ECCV 2014: 13th European Conference, Zurich, Switzerland, September 6-12, 2014, Proceedings, Part V 13 (pp. 740-755). Springer International Publishing. [2] Everingham, M., Van Gool, L., Williams, C.K., Winn, J. and Zisserman, A., 2010. The pascal visual object classes (voc) challenge. International journal of computer vision, 88, pp.303-338. [3] Bai, Y., Geng, X., Mangalam, K., Bar, A., Yuille, A.L., Darrell, T., Malik, J. and Efros, A.A., 2024. Sequential modeling enables scalable learning for large vision models. In Proceedings of the IEEE/CVF Conference on Computer Vision and Pattern Recognition (pp. 22861-22872).

---

### Official Review · Reviewer_CkD8 · 2024-11-04

**Soundness:** 3
**Presentation:** 4
**Contribution:** 3
**Rating:** 5
**Confidence:** 3

**Summary:**

The paper introduces Chain-of-Focus (CoF) Prompting as a method to enhance large autoregressive vision models (LAVMs) for in-context learning (ICL) in computer vision. Drawing inspiration from Chain-of-Thought prompting in natural language processing, CoF Prompting enables vision models to perform step-by-step visual comprehension. It tackles the lack of logical structure in visual data by generating intermediate reasoning steps based on visual saliency. CoF Prompting also facilitates the creation of customized prompts by selecting the most contextually relevant ones, based on query similarity and target richness. The effectiveness of this method is supported by extensive experiments, demonstrating that it significantly improves the models' ability to make inferences on visual understanding tasks, leveraging the recent advancements in LAVMs that utilize purely visual inputs for ICL.

**Strengths:**

- This paper introduces the interesting method of Chain-of-Focus Prompting, with strong and reasonable motivation.

- The experiments are comprehensive.

**Weaknesses:**

- Compared to previous work, such as "A Generalist Painter for In-Context Visual Learning," this paper only addresses two tasks: image segmentation and pose estimation. If this method does not work for other tasks, it could decrease the contribution of the paper.

- I am somewhat skeptical about why this method works. For instance, would the model perform equally well if it viewed the same number of images within reasoning steps (not randomly selected, such as viewing the same number of final step reasoning images)?

[1] Wang, Xinlong, et al. "Images speak in images: A generalist painter for in-context visual learning." Proceedings of the IEEE/CVF Conference on Computer Vision and Pattern Recognition. 2023.

**Questions:**

- Would the model perform equally well if it viewed the same number of images within reasoning steps (not randomly selected, such as viewing the same number of final step reasoning images)?

- If you simply use this logic for Chain-of-Focus: each step of reasoning progresses from images of a single object to two, and then to multiple objects, where do you think the disadvantages lie compared to your method?

---

> ### Author Response · Authors · 2024-11-24
> **Response to Reviewer CkD8 (Part1, Q1)**
>
> >We thank Reviewer CkD8 for the thoughtful feedback and valuable suggestions, which have greatly helped us improve the clarity and depth of our work. Following are the responses to the questions and comments.
>
> ---
>
> > **Q1: View the same number of final step reasoning images**
>
> >A1: Thank you for this insightful question. To investigate this, we first conducted experiments where the original reasoning steps were replaced with repeated instances of the same final step. Below are the results for the four tasks using the 7B LAVM. "w/ FS" represents final steps, while "w/ Chain" corresponds to the original CoF prompting method.
>
> >| Prompting Method\Task  | Segmentation                         | Pose Estimation                      | Object Detection   | Image Inpainting                          |
> |------------------------|--------------------------------------|--------------------------------------|--------------------|-------------------------------------------|
> |                        | IoU ($\uparrow$)/ P-ACC ($\uparrow$) | IoU ($\uparrow$)/ P-ACC ($\uparrow$) | L-IoU ($\uparrow$) | MSE ($\downarrow$) / LPIPS ($\downarrow$) |
> | CoF Prompting w/ FS    | 50.37 / 64.10                        | 2.67 / 11.64                         | 19.66              | 0.88 / 0.57                               |
> | CoF Prompting w/ Chain | 52.53 / 67.05                        | 2.80 / 13.34                         | 19.74              | 0.61 / 0.47                               |
> | Difference             | +2.16 / +2.95                        | +0.13 / +1.70                        | +0.08              | +0.27 / +0.10                              |
>
>
>
> >Our findings indicate that the performance of the method slightly decreases across all four tasks when it views repeated final steps instead of chained reasoning steps. We hypothesize that repeating the final steps may not introduce additional prior knowledge for the autoregressive models, whereas chained reasoning steps provide structured steps that address this limitation effectively.
>
> >We further investigate the scenario where the model observes more final steps without repeating the same prompt pairs, where the model processes a greater number of queries along with their respective final steps. We hypothesize that adding more relevant prompt pairs can enhance a LAVM’s in-context learning performance, similar to including additional details in a language question for Large Language Models (LLMs) can encourage them to produce more accurate responses. The primary focus of our work, however, is on examining whether incorporating intermediate reasoning steps into visual targets can further boost the performance of LAVMs.
>
> >To validate this, we conducted a series of experiments. First, we tested whether using more diverse and relevant visual prompt images improves performance. Next, we evaluated whether adding reasoning steps to these examples provides additional benefits. As shown in the table below, the results support our hypothesis: across all four tasks, there is a linear positive correlation between the number of given prompt pairs and the metric performance. This confirms that incorporating more relevant visual prompt pairs enhances the in-context learning capabilities of the 7B LAVMs.
>
> >We then compared the in-context learning performance using the same prompt images with only one reasoning step. Our findings show that adding intermediate reasoning steps further improves performance. Interestingly, in some cases, the improvement gained by introducing a reasoning step exceeds that of including an additional prompt image.
>
> >These two approaches, using more diverse visual prompts and incorporating intermediate reasoning steps, represent complementary research directions. The former emphasizes expanding and refining the prompt query space to enhance in-context learning, while the latter focuses on extracting information through structured reasoning steps based on the prompt target.
>
> >In this work, we have explored the latter approach and demonstrated that incorporating reasoning steps significantly enhances the in-context predictive ability of LAVMs. Importantly, these two strategies are not mutually exclusive; instead, they complement each other and can jointly improve the in-context learning performance of LAVMs.

---

> ### Author Response · Authors · 2024-11-24
> **Response to Reviewer CkD8 (Part2, Q1 continued and Q2)**
>
> >A1-continued:
>
> >| Prompt Images | Reasoning Steps | Segmentation (IoU / P-ACC) | Pose Estimation (IoU / P-ACC) | Object Detection (L-IoU) | Inpainting (MSE $\downarrow$ / LPIPS $\downarrow$) |
> |---------------|-----------------|----------------------------|-------------------------------|--------------------------|----------------------------------------------------|
> | 1             | N               | 50.29 / 64.60              | 2.66 / 12.87                  | 18.65                    | 0.79 / 0.52                                        |
> | 1             | Y               | 51.08 / 66.17              | 2.76 / 12.29                  | 19.16                    | 0.74 / 0.50                                        |
> | 2             | N               | 50.27 / 64.54              | 2.62 / 11.04                  | 18.92                    | 0.69 / 0.50                                        |
> | 2             | Y               | 51.79 / 66.82              | 2.78 / 13.27                  | 19.71                    | 0.60 / 0.45                                        |
> | 3             | N               | 52.57 / 68.16              | 2.74 / 12.14                  | 19.25                    | 0.59 / 0.45                                        |
> | 3             | Y               | 54.19 /68.73               | 2.81 / 12.87                  | 19.77                    | 0.59 / 0.42                                        |
> | 4             | N               | 54.28 / 68.59              | 2.92 / 13.10                  | 19.73                    | 0.56 / 0.44                                        |
> | 4             | Y               | 54.70 / 69.04              | 2.96 / 13.77                  | 19.86                    | 0.56 / 0.40                                        |
>
> ---
>
> > **Q2: Progresses by one object at a time**
>
> > A2: To address this question, we first clarify the differences between CoF prompting and the suggested strategy. The differences are twofold: (1) the step size in our reasoning process is larger (more than one object at a time, depending on the total number of objects and the number of reasoning steps), and (2) CoF prompting specifies the order in which objects are presented during prompting.
>
> > For the first point, we believe that a smaller step size of one object at a time could also work within our framework. In fact, this approach could provide the model with more fine-grained reasoning details. However, depending on the total number of objects in the visual input, the resulting reasoning chain could become very large, potentially exceeding the capacity of current LAVMs (i.e., surpassing the maximum number of tokens in the input sequence). To validate our hypothesis regarding better reasoning performance with a smaller step size, we conducted an experiment on a subset of validation data containing fewer than 12 objects. This ensures that the generated reasoning chain remains within the capacity of the current pre-trained LAVMs. We adapted the "one object at a step" strategy, and the results, presented in the table below, show that stepping one object at a time yields better performance than the original method in segmentation and detection tasks. This indicates that the strategy works well within our framework. However, adopting this method imposes a significant limitation: for visual inputs with many objects, this approach generates a prompt set that exceeds the capacity of current pre-trained LAVMs, restricting the applicability of the prompting method.
>
> > For the second point, not specifying the order of selected objects has the advantage of reducing the computational cost of calculating saliency scores, making it more time and resource efficient. However, the order of presenting selected objects can impact the in-context learning performance. For instance, if the reasoning steps begin with non-salient objects, the visualized target may become sparse and contain redundant information (large areas of black background) which might hinder the model's ability to learn effectively. In contrast, CoF prompting employs a structured approach, presenting objects based on their saliency. This mitigates redundancy and ensures that the focus of the LAVMs is guided appropriately during in-context predictions. Our experiments, as shown in Table 8 and Figure 7, demonstrate an extreme case of the suggested strategy: reversing the reasoning steps in CoF prompting. In this scenario, non-salient objects are presented in the intermediate steps first, resulting in a noticeable decline in the LAVMs' in-context predictive performance. These findings highlight the importance of structured reasoning in CoF prompting for achieving optimal performance.

---

> ### Author Response · Authors · 2024-11-24
> **Response to Reviewer CkD8 (Part3, Q2 continued and Q3)**
>
> > A2-continued:
>
> > | Prompting Method\Task         | Segmentation                         | Pose Estimation                      | Object Detection   | Image Inpainting                          |
> |-------------------------------|--------------------------------------|--------------------------------------|--------------------|-------------------------------------------|
> |                               | IoU ($\uparrow$)/ P-ACC ($\uparrow$) | IoU ($\uparrow$)/ P-ACC ($\uparrow$) | L-IoU ($\uparrow$) | MSE ($\downarrow$) / LPIPS ($\downarrow$) |
> | CoF Prompting w/ Random One Step | 50.74 / 66.32                        | 2.74 / 11.59                          | 19.95              | 0.80 / 0.59                               |
> | CoF Prompting w/ One Step        | 56.32 / 69.86                        | 2.84 / 13.42                         | 22.30              | 0.61 / 0.48                               |
> | CoF Prompting                 | 52.94 / 67.28                        | 2.91 / 13.75                         | 21.52              | 0.60 / 0.46                               |
>
> ---
>
> > **Q3: Task addressed**
>
> > A3:  We would like to kindly clarify that our paper addresses 4 tasks: image segmentation, pose estimation, object detection, and image inpainting. Due to page constraints, the quantitative analysis for object detection and image inpainting is provided in the appendix. The results demonstrate that the proposed method works effectively across all tasks, improving the in-context predictive performance of LAVMs.
>
> > We realized that the references to these sections in the appendix were not prominently marked. To address this, we have revised the relevant sections of the paper, clearly highlighting the tasks and their corresponding sections in bold for better readability.
>
> ---
>
> **We hope these answers have addressed all your concerns. Please kindly let us know if you have any further questions.**

---

> ### Author Response · Authors · 2024-11-25
>
> Dear reviewer CkD8,
>
> Thank you for your time and constuctive comments! This is a gentle reminder that the discussion period is nearing its conclusion, we hope you have taken the time to consider our responses to your review. If you have any additional questions or concerns, please let us know so we can resolve them before the discussion period concludes. If you feel our responses have satisfactorily addressed your concerns, it would be greatly appreciated if you could raise your score to show that the existing concerns have been addressed.
>
> Thank you!
> The Authors

---

> > ### Author Response · Authors · 2024-11-28
> >
> > Dear Reviewer ckD8,
> >
> > This is a warm reminder that we are still waiting for your reply.
> >
> >
> > Many thanks,
> >
> > Authors

---

> > > ### Author Response · Authors · 2024-12-02
> > > **A kind reminder**
> > >
> > > Dear Reviewer ckD8,
> > >
> > > We have tried our best to follow your contuctive comments and revised our paper. Thank you very much for your help.
> > >
> > > As the discussion period is nearing its conclusion, could you please kindly let us know if there are any remaining concerns. We are more than happy to provide a response.
> > >
> > > Many thanks,
> > >
> > > Authors

---

### Meta-Review · Area_Chair_Xq1L · 2024-12-19

**Metareview:**

This paper introduces chain-of-focus prompting, a new approach that enhances vision models through step-by-step visual comprehension and contextually informative prompts, enabling better inferences in visual in-context learning tasks. The proposed method has strong and reasonable motivation. The experiments are comprehensive.

The reviewers raised concerns about data privacy, resource overhead, limited exploration of computational complexity, and the narrow focus on segmentation and pose estimation, suggesting broader task evaluation and deeper analysis of failure cases. The authors provided detailed responses. Three reviewers of 4 have expressed the satisfication of the responses. Reviewer CkD8 did not further response. I think the response has well addressed the concern of CkD8.

Given the overall score of 6.75 and concerns beening addressed, AC suggest to accept this paper.

**Additional Comments On Reviewer Discussion:**

The reviewers raised concerns about data privacy, resource overhead, limited exploration of computational complexity, and the narrow focus on segmentation and pose estimation, suggesting broader task evaluation and deeper analysis of failure cases. The authors provided detailed responses. Three reviewers of 4 have expressed the satisfication of the responses. Reviewer CkD8 did not further response. I think the response has well addressed the concern of CkD8.

---

### Decision · Program_Chairs · 2025-01-22

Accept (Poster)